# CO³GESTURE: TOWARDS COHERENT CONCURRENT CO-SPEECH 3D GESTURE GENERATION WITH INTERACTIVE DIFFUSION

**Xingqun Qi**[1,*], **Yatian Wang**[1,*], **Hengyuan Zhang**[2], **Jiahao Pan**[1]
**Wei Xue**[1], **Shanghang Zhang**[2], **Wenhan Luo**[1], **Qifeng Liu**[1,✉], **Yike Guo**[1,✉]
[1] The Hong Kong University of Science and Technology
[2] Peking University

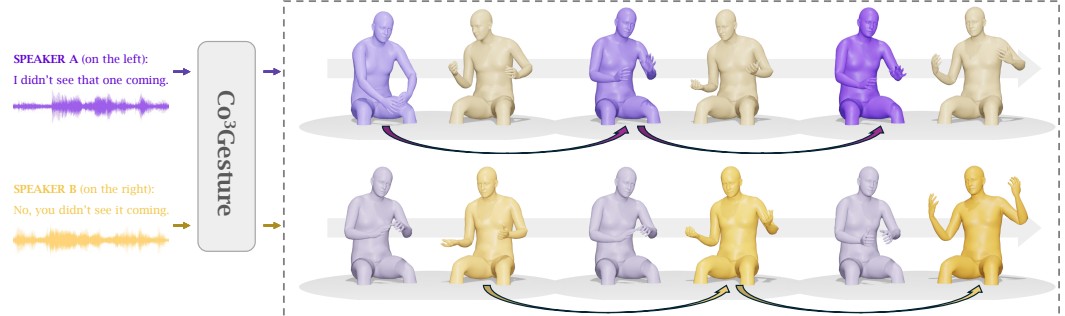

Figure 1: Diverse exemplary clips sampled by our method from our newly collected **GES-Inter dataset**. The vital frames are visualized to demonstrate the concurrent upper body dynamics of two speakers generated by our **Co³Gesture** framework displaying temporal coherent interaction with each other, respectively. Best view on screen.

## ABSTRACT

Generating gestures from human speech has gained tremendous progress in animating virtual avatars. While the existing methods enable synthesizing gestures cooperated by individual self-talking, they overlook the practicality of concurrent gesture modeling with two-person interactive conversations. Moreover, the lack of high-quality datasets with concurrent co-speech gestures also limits handling this issue. To fulfill this goal, we first construct a large-scale concurrent co-speech gesture dataset that contains more than 7M frames for diverse two-person interactive posture sequences, dubbed **GES-Inter**. Additionally, we propose **Co³Gesture**, a novel framework that enables coherent concurrent co-speech gesture synthesis including two-person interactive movements. Considering the asymmetric body dynamics of two speakers, our framework is built upon two cooperative generation branches conditioned on separated speaker audio. Specifically, to enhance the coordination of human postures *w.r.t.* corresponding speaker audios while interacting with the conversational partner, we present a Temporal Interaction Module (**TIM**). TIM can effectively model the temporal association representation between two speakers' gesture sequences as interaction guidance and fuse it into the concurrent gesture generation. Then, we devise a mutual attention mechanism to further holistically boost learning dependencies of interacted concurrent motions, thereby enabling us to generate vivid and coherent gestures. Extensive experiments demonstrate that our method outperforms the state-of-the-art models on our newly collected GES-Inter dataset. The dataset and source code are publicly available at *https://mattie-e.github.io/Co3/*.

---

[*]These authors contributed equally to this work.
✉ Corresponding authors.

# 1 INTRODUCTION

The generation of co-speech gestures seeks to create expressive and diverse human postures that align with audio input. These non-verbal behaviors play a crucial role in human communication, significantly enhancing the effectiveness of speech delivery. Meanwhile, modeling co-speech gestures has broad applications in embodied AI, including human-machine interaction (Liu et al., 2023), robotic assistance (Farouk, 2022), and virtual/augmented reality (AR/VR) (Fu et al., 2022). Traditionally, researchers have primarily concentrated on synthesizing upper body gestures that correspond to spoken audio (Liu et al., 2022b; Yi et al., 2023).

These methods usually focus on synthesizing single-speaker gestures following people's self-talking (Liu et al., 2024b;a; Qi et al., 2024b; Yang et al., 2023). Although some researchers model the single human postures via conversational speech corpus (Mughal et al., 2024a; Ng et al., 2024), they mostly overlook generating the concurrent long sequence gestures with interactions. Besides, others generate the single speaker gesture from conversational corpus incorporated with interlocuter reaction movements (Kucherenko et al., 2023; Zhao et al., 2023). However, few researchers have devoted themselves to constructing datasets with interactive concurrent gestures. For example, the interactive movement of two speakers may include waving arms when saying "*hello*" during conversation. In this work, we therefore introduce the new task of two-speaker concurrent gestures generation under the condition of conversational human speeches, as displayed in Figure 1.

There are two main challenges in this task: 1) Datasets of concurrent 3D co-speech gestures synchronized with conversation audios of two speakers are scarce. Creating such a dataset containing large-scale 3D human postures is difficult due to complex motion capture systems and expensive labor for actors. 2) Modeling the plausible and temporal coherent co-speech gestures of two speakers is difficult, especially involving the frequent interactions in long sequences.

To address the issue of data scarcity, we construct a new large-scale whole-body meshed 3D co-speech gesture dataset that includes concurrent speaker postures within more than 7M frames, dubbed **GES-Inter**. In particular, we first leverage the advanced 3D pose estimator (Zhang et al., 2023a) to obtain high-quality poses (*i.e.*, SMPL-X Pavlakos et al. (2019) and FLAME Li et al. (2017)) from in-the-wild talk show videos. To obtain the individual sound signals of each speaker in the conversation while preserving the identity consistency with the posture movement, we employ the pyannote-audio Bredin et al. (2020) to separate the mixed speech, as shown in Figure 2. Afterward, by utilizing the automatic speech recognition techniques Whisper-X Bain et al. (2023), we acquire the consistent text transcript and speech phoneme with speaker audio. In this fashion, our GES-Inter dataset covers a wide range of two-person interactive concurrent co-speech gestures, from daily conversations to formal interviews. Moreover, the multi-modality

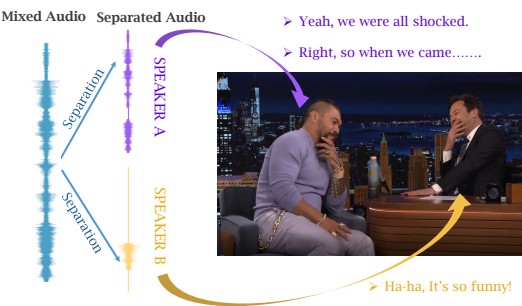

Figure 2: Illustration of our audio separation and alignment with speakers.

annotation and common meshed human postures pave the potential for various downstream tasks like human behaviors analysis (Liang et al., 2024b; Xu et al., 2024) and talking face generation (Peng et al., 2024; Ng et al., 2024), *etc*.

Based on our GES-Inter dataset, we propose a novel framework, named **Co³Gesture** , to model the coherent concurrent co-speech 3D gesture generation. The key insight of our framework is to carefully build the interactions between concurrent gestures. Here, we observe that the motions of two speakers are asymmetric (*e.g.*, when one speaker moves in talking, the other could be quiet in static or moving slowly). Directly producing the concurrent gestures in a holistic manner may lead to unnatural results. Therefore, we establish two cooperative transformer-based diffusion branches to generate corresponding gestures of two speakers, performing the specific denoising process, respectively. This bilateral paradigm encourages our framework to yield diverse interactive movements while effectively preventing mode collapse.

Moreover, to ensure the motions of the one speaker are temporally consistent with the corresponding audio signal and display coherent interaction with the conversational partner, we devise a Temporal Interaction Module (**TIM**). Specifically, we first incorporate the separated human voices to produce single-speaker gesture features, respectively. Then, we model the joint embedding of the current speaker features and the integrated conversational motion clues guided by mixed speech audio. Here, the learned joint embedding is leveraged as the soft weight to balance the interaction dependence of the generated current speaker gesture dynamics with other ones. Then, we conduct the mutual attention of the fused bilateral gesture denoisers to further facilitate high-fidelity concurrent gesture generation with desirable interactive properties. Extensive experiments conducted on our newly collected GES-Inter dataset verify the effectiveness of our method, displaying diverse and vivid concurrent co-speech gestures.

Overall, our contributions are summarized as follows:

- We introduce the new task of concurrent co-speech gesture generation cooperating with one newly collected large-scale dataset named GES-Inter. It contains more than 7M high-quality co-speech postures of the whole body, significantly facilitating research on diverse gesture generation.

- We propose a novel framework named Co$^3$Gesture upon the bilateral cooperative diffusion branches to produce realistic concurrent co-speech gestures. Our Co$^3$Gesture includes the tailor-designed Temporal Interaction Module (TIM) to ensure the temporal synchronization of gestures *w.r.t.* the corresponding speaker voices while preserving desirable interactive dynamics.

- Extensive experiments show that our framework outperforms various state-of-the-art counterparts on the GES-Inter dataset, producing diverse and coherent concurrent co-speech gestures given conversational speech audios.

## 2   RELATED WORK

**Co-speech Gesture Synthesis.**   Synthesizing the diverse and impressive co-speech gestures displays a significant role in the wide range of applications like human-machine interaction (Cho et al., 2023; Guo et al., 2021), robot (De Wit et al., 2023; Sahoo et al., 2023), and embodied AI (Li et al., 2023; Benson et al., 2023). Numerous works were proposed to address this task that can be roughly divided into rule-based approaches, machine learning designed methods, and deep learning based ones. Rule-based research depends on linguistic experts' pre-defined corpus to bridge human speech and gesture movements (Cassell et al., 1994; Poggi et al., 2005). The others usually leverage machine learning techniques with mutually constructed speech features to generate co-speech gestures (Levine et al., 2010; Sargin et al., 2008). However, these methods heavily rely on efforts on pre-processing which may cause expensive labor consumption.

Recently deep learning based methods gained much development directly modeling co-speech gesture synthesis via deep neural networks. Most of them usually leverage the multi-modality cues to generate postures incorporated with individual self-talking audio (Li et al., 2021a; Zhu et al., 2023; Yi et al., 2023; Qi et al., 2024c), such as speaker identity (Liu et al., 2024a;b; Chen et al., 2024), emotion (Qi et al., 2024b;a; Liu et al., 2022a), and transcripts (Zhang et al., 2024; Ao et al., 2023; 2022; Zhi et al., 2023). Only a few counterparts propose to synthesize the single gesture under conversational speech guidance (Ng et al., 2024; Mughal et al., 2024b). Besides, the GENEA challenge holds the most similar settings to us. The participants aim to generate the single-person gesture from the conversational corpus incorporated with interlocuter reaction movements (Kucherenko et al., 2023). However, they overlook the concurrent co-speech gesture modeling of two speakers during the conversation is much more practical in the real scenes. Few of the above methods could be directly adapted to this new thought.

**Co-speech Gesture Datasets.**   Co-speech gesture datasets are roughly divided into two types: pseudo-label based gestures and motion-capture based ones. For pseudo-label approaches, researchers usually utilize the pre-trained pose estimator to obtain upper body postures from in-the-wild News or talk show videos (Yoon et al., 2020; Ahuja et al., 2020; Habibie et al., 2021). Thanks to the recent advanced parametric whole-body meshed 3D model SMPL-X Pavlakos et al. (2019) and FLAME Li et al. (2017), some high-quality whole-body based 3D co-speech gesture datasets are emerged (Qi et al., 2024c; Yi et al., 2023; Qi et al., 2024b;a). Meanwhile, it significantly promotes the construction

Table 1: Statistical comparison of our GES-Inter with existing datasets. The dotted line separates whether the speech content in the dataset is built based on the conversational corpus.

| Datasets | Concurrent Gestures | Duration (hours) | Additional Attributes | | | | Joint Annotation |
|---|---|---|---|---|---|---|---|
| | | | Facial | Mesh | Phonme | Text | |
| TED (Yoon et al., 2020)$_{TOG}$ | ✗ | 106.1 | ✗ | ✗ | ✗ | ✓ | pseudo |
| TED-Ex (Liu et al., 2022b)$_{CVPR}$ | ✗ | 100.8 | ✗ | ✗ | ✗ | ✓ | pseudo |
| EGGS (Ghorbani et al., 2023)$_{CGF}$ | ✗ | 2 | ✗ | ✗ | ✗ | ✓ | pseudo |
| BEAT (Liu et al., 2022a)$_{ECCV}$ | ✗ | 76 | ✓ | ✗ | ✓ | ✓ | mo-cap |
| SHOW (Yi et al., 2023)$_{CVPR}$ | ✗ | 26.9 | ✓ | ✓ | ✗ | ✗ | pseudo |
| TWH16.2 (Lee et al., 2019)$_{ICCV}$ | ✓ | 17 | ✓ | ✓ | ✓ | ✓ | mo-cap |
| BEAT2 (Liu et al., 2024a)$_{CVPR}$ | ✗ | 60 | ✓ | ✓ | ✓ | ✓ | mo-cap |
| DND (Mughal et al., 2024b)$_{CVPR}$ | ✗ | 6 | ✗ | ✗ | ✗ | ✗ | mo-cap |
| Photoreal (Ng et al., 2024)$_{CVPR}$ | ✗ | 8 | ✓ | ✗ | ✗ | ✗ | mo-cap |
| **GES-Inter (ours)** | ✓ | **70** | ✓ | ✓ | ✓ | ✓ | **pseudo** |

of motion-capture based co-speech datasets (Liu et al., 2022a; 2024a; Ghorbani et al., 2023; Mughal et al., 2024b; Ng et al., 2024). Although some of them are built upon conversational corpora, they only provide gestures of single speakers (Liu et al., 2024a; Ng et al., 2024). The TWH16.2 (Lee et al., 2019) dataset displays the pioneer exploration of concurrent gestures via joint-based representation. However, it overlooks the significance of the facial expression data in conversation. Meanwhile, the SMPL-X mesh-based whole-body data in our dataset is more convenient for avatar rendering and downstream applications (*e.g.*, talking face) compared to TWH16.2. Besides, the DND GROUP GESTURE dataset Mughal et al. (2024b) is built upon a multi-performer group talking scene, which can not be directly applied to our task. Therefore, a 3D co-speech dataset including concurrent gestures of two speakers with the meshed whole body is required for further research.

**3D Human Motion Modeling.** Human motion modeling aims to generate natural and realistic coherent posture sequences from multi-modality conditions, which contains co-speech gesture synthesis as a sub-task (Liang et al., 2024a; Tevet et al., 2023). One of the hottest tasks is generating human movements from the input action descriptions (Jiang et al., 2023; Zhang et al., 2023b; Lin et al., 2024; Xu et al., 2024). It needs to enforce the results by displaying an accurate semantic expression aligned with text prompts. The other one that shares modality guidance similar to our task is the AI choreographer (Li et al., 2020; 2021b; Siyao et al., 2022; Le et al., 2023). While retaining analogous interactive human motion modeling with the approaches mentioned above, our newly introduced work differs from them significantly. Both of the aforementioned topics follow the symmetrical fact that exchanging the identities of performers during interactions does not change the semantics or coherence of motions. We take the asymmetric body dynamics of concurrent human movements into consideration, motivating us to design the bilateral diffusion branches.

## 3 PROPOSED METHOD

### 3.1 INTERACTIVE GESTURE DATASET CONSTRUCTION

**Preliminary.** Due to the expensive labor and complex motion capture system establishment during the frequent interactive conversations, similar to Yi et al. (2023); Liu et al. (2022b); Qi et al. (2024c), we intend to obtain the high-quality 3D pseudo human postures of our dataset. Synthesizing datasets conducive to our task focuses on ensuring high-fidelity and smooth gesture movements, authority speaker voice separation, and identity consistent audio-posture alignment.

**Estimation of 3D Posture.** Firstly, we exploit the state-of-the-art 3D pose estimator Pymaf-X Zhang et al. (2023a) to acquire the meshed whole-body parametric human postures based on SMPL-X Pavlakos et al. (2019). In particular, the body dynamics are denoted by the unified SMPL model Loper et al. (2023) that collaborated with the MANO hand model Boukhayma et al. (2019). Meanwhile, we adopt the FLAME face model Li et al. (2017) to present the facial expressions of speakers. The corpora are collected from the in-the-wild talk show or formal interview videos that contain high-resolution frames and unobstructed sitting postures. Then, we conduct extensive

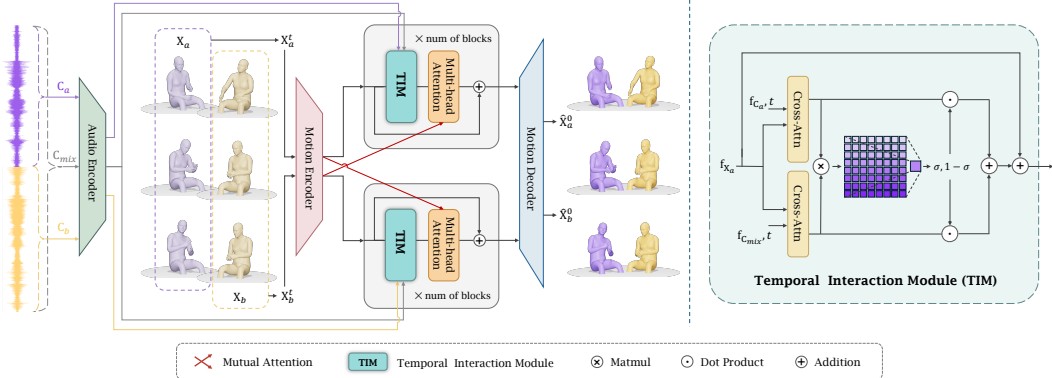

Figure 3: The overall pipeline of our Co$^3$Gesture . Given conversational speech audios, our framework generates concurrent co-speech gestures with coherent interactions.

data processing to filter the unnatural and jittery poses, thereby ensuring the high-quality of the dataset[*]. Our GES-Inter includes more than 7M validated gesture frames with 70 hours. To the best of our knowledge, this is the first large-scale co-speech dataset that includes mesh-based whole-body concurrent postures of two speakers, as reported in Table 1.

**Separation of Speaker Audio.** To obtain the identity-specific speech audios of each speaker in the mixed conversation corpus, we leverage the advanced sound source separation technique pyannote-audio Bredin et al. (2020) to conduct human voice separation. Here, we enforce the number of separated speakers as two for assigning each speech segment to the corresponding speaker. Then, we utilize the superior speech recognition model WhisperX Bain et al. (2023) to acquire accurate word-level text transcripts. Once we acquire high-fidelity transcripts, we utilize the Montreal Forced Aligner (MFA) McAuliffe et al. (2017) to obtain phoneme-level timestamps associated with facial expressions. Such extensive multi-modality attributes of our dataset enable the research of various downstream tasks like talking face (Peng et al., 2024; Ng et al., 2024), and human behavior analysis (Park et al., 2023; Qi et al., 2023; Dubois et al., 2024) *etc.*

**Alignment of Audio-Posture Pair.** Once we obtain the separated audio signals for each speaker, we assign them to the corresponding body dynamics. We recruit professional inspectors to manually annotate the separated audio signals with their corresponding speaker identities. To ensure the accuracy of the aligned audio-posture pairs, different inspectors double-check these results. In this way, our newly constructed GES-Inter dataset offers high-quality concurrent gestures with aligned multi-modal annotations.

## 3.2 PROBLEM FORMULATION

Given a sequential collection of conversational audio signal $\mathbf{C}_{mix}$ as the condition, our goal is to generate the interactive concurrent gesture sequences of two speakers $\mathbf{x}$. Where $\mathbf{C}_{mix} = \mathrm{C}_a + \mathrm{C}_b$ and $\mathbf{x} = \{\mathrm{x}_a, \mathrm{x}_b\}$ denote the corresponding audios and postures of two speakers. The sequence length is the fixed number $N$. Specifically, each pose of a single person is presented as $J$ joints with 3D representation. Note that we only generate the upper body including fingers in this work.

## 3.3 BILATERAL COOPERATIVE DIFFUSION MODEL

Considering the asymmetric body dynamics of two speakers, we aim to address the concurrent co-speech gesture generation in a bilateral cooperation manner, as depicted in Figure 3. The framework takes the two noisy human motions as input for producing denoised ones, which is conditioned on diffusion time step $t$, mixed conversational audio signal $\mathbf{C}_{mix}$, and separated speaker voices $\mathrm{C}_a$ or $\mathrm{C}_b$. We leverage separated human speech as guidance for bilateral branches to generate

---

[*]Please refer to supplementary material for more details about data processing.

corresponding gestures. Moreover, we utilize the original mixed audio signal of two speakers to indicate the interaction information to ensure the synthesized posture retains rhythm with specific audio while preserving interactive coherency with the conversation partner. All the audio signals are fed into the audio encoder for feature extraction.

**Temporal Interaction Module.** To ensure temporal consistency while preserving the interactive dynamics of concurrent gestures, we propose a Temporal Interaction Module (TIM) to model the temporal association representation between each current speaker movements and conversational counterparts. As shown in Figure 3, we utilize the features extracted from mixed conversational audios to indicate the interaction information. Here, the dimensions of features in our TIM are all normalized as $\mathbb{R}^{N \times D}$. For notation simplicity, we take single-branch $x_a$ for elaboration.

In particular, we first incorporate the current speaker audio embedding $\mathbf{f}_{C_a}$ as the query $Q$ to match the key feature $K$ and value feature $V$ belonging to motion embedding $\mathbf{f}_{x_a}$ via cross-attention meshanism (Vaswani et al., 2017):

$$Q = \mathbf{f}_{C_a} \mathbf{W}, K = \mathbf{f}_{x_a} \mathbf{W}, V = \mathbf{f}_{x_a} \mathbf{W}. \tag{1}$$

Here, $\mathbf{W}$ denotes the projection matrix. Along with this operation, we obtain the updated current speaker motion embedding $\mathbf{f}_{x_a, C_a}$. Similarly, we acquire the interactive motion embedding $\mathbf{f}_{x_a, C_{mix}}$ incorporated with mixed conversational speeches. Then we calculate the temporal correlation matrix $\mathbf{M} \in \mathbb{R}^{N \times N}$ between the updated current gesture embedding and interactive embedding. Here, the temporal correlation matrix represents the temporal variants between the current gesture sequences and interactive ones. Then, we exploit a motion encoder to acquire a learnable weight parameter $\sigma$ as the temporal-interaction dependency. Once we obtain the weight parameter, the current speaker motion embedding is boosted as follows:

$$\mathbf{f}_{x_a, C_a} = \sigma \odot \mathbf{f}_{x_a, C_a} + (1 - \sigma) \odot \mathbf{f}_{x_a, C_{mix}}, \sigma = sigmoid(\text{Enc}(\mathbf{M})), \tag{2}$$

where $\odot$ is Hadamard product, Enc denotes the motion encoder. The motion embedding of the conversation partner is updated in the same manner. In this fashion, the temporal interaction fidelity of generated gestures is well-preserved.

**Mutual Attention Mechanism.** To further enhance the interaction between two speakers, we construct bilateral cooperative branches that interact with each other to produce concurrent gestures. To be specific, we introduce the mutual attention layers that take the features of the counterpart as the query $Q$ in Multi-Head Attention (MHA), respectively. We observe that exchanging the input order of the speaker's audio results in an invariance effect of interactive body dynamics. In other words, the distribution of interaction data of two speakers adheres to the same marginal distribution. Therefore, we formulate the cooperating denoisers retaining shared weight update strategies. This encourages the gesture features after the TIM to be more temporal and interactive with partner ones, holistically.

### 3.4 OBJECTIVE FUNCTIONS

During the training phase, the denoisers of our bilateral branches share the common network structure. Given the diffusion time step $t$, the current speaker audio $\{C_a, C_b\}$, the mixed conversation audio $\mathbf{C}_{mix}$, and the noised gestures $\left\{x_a^{(t)}, x_b^{(t)}\right\}$, the denoisers are enforced to produce continuous human gestures. The denoising process can be constrained by the simple objective:

$$\mathcal{L}_{simple} = \mathbb{E}_{\mathbf{x}, t, \epsilon} \left[ \left\| x_a - \mathcal{D}(x_a^{(t)}, C_a, \mathbf{C}_{mix}, t) \right\|_2^2 + \left\| x_b - \mathcal{D}(x_b^{(t)}, C_b, \mathbf{C}_{mix}, t) \right\|_2^2 \right], \tag{3}$$

where $\mathcal{D}$ is the denoiser, $\epsilon \sim \mathcal{N}(\mathbf{0}, \mathbf{I})$ is the added random Gaussian noise, $x_{\{a,b\}}^{(t)} = x_{\{a,b\}} + \sigma_{(t)}\epsilon$ is the gradually noise adding process at step $t$. $\sigma_{(t)} \in (0, 1)$ is the constant hper-parameter. Moreover, we utilize the velocity loss $\mathcal{L}_{vel}$ and foot contact loss $\mathcal{L}_{foot}$ (Tevet et al., 2023) to provide supervision on the smoothness and physical reasonableness, respectively. Finally, the overall objective is:

$$\mathcal{L}_{total} = \lambda_{simple}\mathcal{L}_{simple} + \mathcal{L}_{vel} + \mathcal{L}_{foot}, \tag{4}$$

where $\lambda_{simple}$ is trade-off weight coefficients.

In the inference, since the audio signals of the concurrent gestures generation serve as an essential condition modality, the prediction of human postures is formulated as fully conditioned denoising. This encourages our framework to strike a balance between high fidelity and diversity.

# 4 EXPERIMENTS

## 4.1 DATASETS AND EXPERIMENTAL SETTING

**GES-Inter Dataset.** Since the existing co-speech gesture datasets fail to provide interactive concurrent body dynamics, we contribute a new dataset named GES-Inter to evaluate our approach. The human postures of our GES-Inter are collected from 1,462 processed videos including talk shows and formal interviews. The extraction takes 8 NVIDIA RTX 4090 GPUs in one month, obtaining 20 million raw frames. After the complex data processing, we get more than 7 million validated instances. Finally, we acquire 27,390 motion clips that are split into training/ validation/ testing following criteria (Liu et al., 2022a; 2024a) as 85%/ 7.5%/ 7.5%.

**Implementation Details.** We set the total generated sequence length $N = 90$ with the FPS normalized as 15 in the experiments. $\mathbf{C}_{mix}$, $C_a$, and $C_b$ are represented as audio signal waves, initially. Then, these audio signals are converted into mel-spectrograms with an FFT window size of 1,024, and the hop length is 512. The dimension of input audio mel-spectrograms is $128 \times 186$. We follow the tradition of (Liu et al., 2022b; Qi et al., 2024b;c) to leverage the speech recognizer as the audio encoder. Each branch of our pipeline is implemented with 8 blocks within 8 heads of attention layers. The latent dimension $D$ is set to 768.

In the training stage, we set $\lambda_{simple} = 15$, empirically. The initial learning rate is set as $1 \times 10^{-4}$ with an AdamW optimizer. Similar to Nichol & Dhariwal (2021), we set the diffusion time step as 1,000 with the cosine noisy schedule. Our model is applied on a single NVIDIA H800 GPU with a batch size of 128. The training takes a total of 100 epochs, accounting for 3 days. During inference, we adopt DDIM Song et al. (2020) sampling strategy with 50 denoising timesteps to produce gestures. Our experiments only contain upper body joints without facial expressions and shape parameters. Our Co³Gesture synthesizes upper body movements containing 46 joints (*i.e.*, 16 body joints + 30 hand joints) of each speaker. Each joint is converted to a 6D rotation representation Zhou et al. for more stable modeling. The dimension of the generated motion sequence is $\mathbb{R}^{90 \times 276}$, where 90 denotes frame number and $276 = 46 \times 6$ means upper body joints. The order of each joint follows the original convention of SMPL-X.

**Evaluation Metrics.** To fully evaluate the realism and diversity of the generated co-speech gestures, we introduce various metrics:

- **FGD**: Fréchet Gesture Distance (FGD) Yoon et al. (2020) is calculated as the distribution distance between the body movements of synthesized results and real ones via a pre-trained autoencoder.

- **BC**: Beat Consistent Score (BC) / Beat Alignment Score (BA) Liu et al. (2022a; 2024a) measures whether the generated motion dynamics are rhythmic consistent with the input speech audios. We report the average score of two speakers in our experiments.

- **Diversity**: Similar to (Liu et al., 2022b; Zhu et al., 2023; Liu et al., 2024a), the autoencoder of FGD is exploited to acquire feature embeddings of the synthesized gestures. Here, the diversity score means the average distance of 500 randomly assembed pairs.

## 4.2 QUANTITATIVE RESULTS

**Comparisons with SOTA Methods.** To the best of our knowledge, we are the first to explore the coherent concurrent co-speech gesture generation with conversational human audio. To fully verify the superiority of our method, we implement various state-of-the-art (SOTA) counterparts from the perspective of single-person-based co-speech gesture generation (*i.e.*, TalkSHOW (Yi et al., 2023), ProbTalk (Liu et al., 2024b), DiffSHEG (Chen et al., 2024), EMAGE (Liu et al., 2024a)) and text2motion generation (*i.e.*, MDM (Tevet et al., 2023), InterX (Xu et al., 2024) InterGen (Liang et al., 2024b)). For fair comparisons, all the competitors are implemented by official source codes or pre-trained models released by authors. Specifically, in DiffSHEG, we follow the convention of the original work to utilize the pre-trained HuBERT (Hsu et al., 2021) for audio feature extraction. In TalkSHOW, we exploit the pre-trained Wav2vec (Baevski et al., 2020) to encode the audio signals following the original setting. Apart from this, the remaining components for gesture generation in DiffSHEG and TalkSHOW are all trained from scratch on the newly collected GES-Inter dataset. For other methods, we modify their final output layer to match the dimensions of our experimental

Table 2: Comparison with the state-of-the-art counterparts on our newly collected GES-Inter dataset. ↑ means the higher the better, and ↓ indicates the lower the better. ± means 95% confidence interval. The dotted line separates whether the methods are adopted from single-person co-speech generation or text2motion counterparts.

| Methods | GES-Inter Dataset | | |
| --- | --- | --- | --- |
| | FGD ↓ | BC ↑ | Diversity ↑ |
| TalkSHOW (Yi et al., 2023)$_{CVPR}$ | 2.256 | 0.613 | $53.037^{\pm 1.021}$ |
| ProbTalk (Liu et al., 2024b)$_{CVPR}$ | 1.238 | 0.645 | $46.981^{\pm 2.173}$ |
| DiffSHEG (Chen et al., 2024)$_{CVPR}$ | 1.209 | 0.638 | $56.781^{\pm 1.905}$ |
| EMAGE (Liu et al., 2024a)$_{CVPR}$ | 1.884 | 0.637 | $60.917^{\pm 1.179}$ |
| MDM (Tevet et al., 2023)$_{ICLR}$ | 1.696 | 0.654 | $65.529^{\pm 2.218}$ |
| InterX (Xu et al., 2024)$_{CVPR}$ | 1.178 | 0.661 | $65.161^{\pm 1.010}$ |
| InterGen (Liang et al., 2024b)$_{IJCV}$ | 1.012 | 0.670 | $69.455^{\pm 1.590}$ |
| **Co$^3$Gesture (ours)** | **0.769** | **0.692** | $\mathbf{72.824}^{\pm 2.026}$ |

settings. Since the above text2motion counterparts are designed without the audio incorporation setting, we adopt the same audio encoder as ours in the models.

As reported in Table 2, we adopt the FGD, BC, and diversity for a well-rounded view of comparison. Our Co$^3$Gesture outperforms all the competitors by a large margin on the GES-Inter dataset. Remarkably, our method even achieves more than 24% (*i.e.*, $(1.102 - 0.769)/1.012 \approx 24\%$) improvement over the sub-optimal counterpart in FGD. We observe that both InterGen (Liang et al., 2024b) and ours synthesize the authority gestures with much higher diversity than others. This is caused by both of them employing the bilateral branches to generate concurrent gestures of two speakers. However, InterGen shows lower performance on FGD due to the lack of effective temporal interaction modeling. In terms of BC, our method attains much better results than other counterparts. This aligns highly with our insight on the audio separation-based bilateral diffusion backbone that encourages each branch to synthesize speech coherent gestures while preserving the vivid interaction of the results. Compared with single-person co-speech gesture based ones, our model still achieves the best performance. This can be attributed to our well-designed temporal interaction module.

**Ablation Study.** To further evaluate the effectiveness of our Co$^3$Gesture , we conduct extensive ablation studies of different components and experiment settings as variations.

**Effects of the TIM and Mutual Attention:** To verify the effectiveness of TIM and mutual attention mechanism, we conduct detailed experiments as reported in Table 3. The exclusion of the temporal interaction module (TIM) and the mutual attention mechanism lead to performance degradation in our full-version framework, respectively. Moreover, we conduct ablation by simply replacing the TIM with an MLP layer for feature fusion, the FGD and BC display the obvious worse impact as shown in the Table. The results verify that our TIM effectively enhances interactive coherency between two speakers. In particular, our temporal interaction module effectively models the temporal dependency between the gesture motions of the current speaker and the conversation partner. Therefore, implementation without it leads our framework to fail in producing cooperative motions, thus significantly reducing the performance in all the metrics. Moreover, the exclusion of the mutual attention mechanism results in FGD is obviously worse than the full version framework. This indicates that our mutual attention can effectively handle complex interactions from the perspective of holistic fashion while balancing the specific movements of two speakers.

Table 3: Ablation study of TIM and mutual attention mechanism on our GES-Inter dataset.

| Methods | GES-Inter Dataset | | |
| --- | --- | --- | --- |
| | FGD ↓ | BC ↑ | Diversity ↑ |
| w/ MLP | 1.202 | 0.663 | $64.690^{\pm 1.137}$ |
| w/o TIM | 1.297 | 0.676 | $67.953^{\pm 1.203}$ |
| w/o Mutual Attention | 0.924 | 0.681 | $69.084^{\pm 1.412}$ |
| **Co$^3$Gesture (full version)** | **0.769** | **0.692** | $\mathbf{72.824}^{\pm 2.026}$ |

**Effects of the Bilateral Branches and Audio Mixed/ Separation:** We also conduct the ablation study in different experiment settings. To demonstrate the effectiveness of our bilateral diffusion branches, we construct the single branch-based diffusion pipeline that generates the gestures of two speakers in a holistic manner. As shown in Table 4, by subtracting the bilateral branches from the full version pipeline, the indicator FGD displays much worse results (*e.g.*, $0.769 \rightarrow 1.669$). This outcome verifies that our cooperative bilateral diffusion branches effectively handle the asymmetric motion of concurrent gestures of two speakers. This supports our key technical insight on framework construction. Then, by subtracting the original mixed audio, the indicators FGD and BC present much worse performance. These results verify the mixed audio signal displays effectively enhance the interaction between two speakers.

Meanwhile, we further verify the capability of our audio separation design where the model only takes the mixed conversational speech signal as input. Based on the above-mentioned fashion, the BC metric clearly attains a much worse result than the separated one. Directly modeling mixed conversational speech to produce concurrent gestures impacts the interactive correlation between two speakers. This seriously affects the harmony of synthesized concurrent gestures and corresponding speech rhythm.

Table 4: Ablation study of bilateral branches and audio mixed/ separation on our GES-Inter dataset.

| Methods | GES-Inter Dataset | | |
|---|---|---|---|
| | FGD $\downarrow$ | BC $\uparrow$ | Diversity $\uparrow$ |
| w/o Bilateral Branches | 1.669 | 0.640 | $64.542^{\pm 1.252}$ |
| w/o Mixed Audio | 1.227 | 0.656 | $64.899^{\pm 1.004}$ |
| w/o Audio Separation | 1.180 | 0.633 | $66.159^{\pm 1.501}$ |
| **Co$^3$Gesture (full version)** | **0.769** | **0.692** | **$72.824^{\pm 2.026}$** |

**Effects of the Foot Contact Loss:** Inspired by (Tevet et al., 2023; Liang et al., 2024b), we introduce foot contact loss to ensure the physical reasonableness of the generated gestures.

Since we only model the upper body joints in experiments, we complete the lower body joints as $T$ pose in forward kinematic function during calculate loss. We conduct the ablation study to verify the effectiveness of $\mathcal{L}_{foot}$. As illustrated in Table 5, the exclusion of the foot contact loss results in FGD and BC are obviously worse than the full version framework. This indicates that our foot contact loss displays a positive impact on the generated postures.

Table 5: Ablation study of foot contact loss on our GES-Inter dataset.

| Methods | GES-Inter Dataset | | |
|---|---|---|---|
| | FGD $\downarrow$ | BC $\uparrow$ | Diversity $\uparrow$ |
| w/o Foot Contact $\mathcal{L}_{foot}$ | 1.082 | 0.675 | $68.448^{\pm 1.082}$ |
| **Co$^3$Gesture (full version)** | **0.769** | **0.692** | **$72.824^{\pm 2.026}$** |

## 4.3 QUALITATIVE EVALUATION

**Visualization.** To fully demonstrate the superior performance of our method, we display the visualized keyframes generated by our Co$^3$Gesture framework with other counterparts, as illustrated in Figure 4. For better demonstration, the relative position coordinates of the two speakers are fixed in visualization. The lower body including the legs is fixed (*e.g.*, seated) while visualizing due to the weak correlation with human speech. For example, it is quite challenging to model whether the two speakers are sitting or standing from only audio inputs. We showcase the two optimal methods from single-person gesture generation and text2motion, respectively. Our method shows coherent and interactive body movements against other ones. To be specific, we observe that ProbTalk and DiffSHEG would synthesize the stiff results (*e.g.*, blue rectangles of right speakers). Although the Inter-X generates the natural movements of the left speaker, it displays less interactive dynamics of the right speaker. In addition, the results synthesized by InterGen show reasonable interaction between two speakers. However, it may produce unnatural postures sometimes (as depicted in red circles). In contrast, our Co$^3$Gesture can generate interaction coherent concurrent co-speech gestures. This highly aligns with our insight into the bilateral cooperative diffusion pipeline. For more visualization demo videos please refer to our anonymous webpage: *https://mattie-e.github.io/Co3/*. In our experiments, the length of the generated gesture sequence is 90 frames with 15 FPS. Thus, all the demo videos in the user study have the same length of 6 seconds.

**User Study.** To further analyze the quality of concurrent gestures synthesized by ours against various competitors, we conduct a user study by recruiting 15 volunteers. All the volunteers

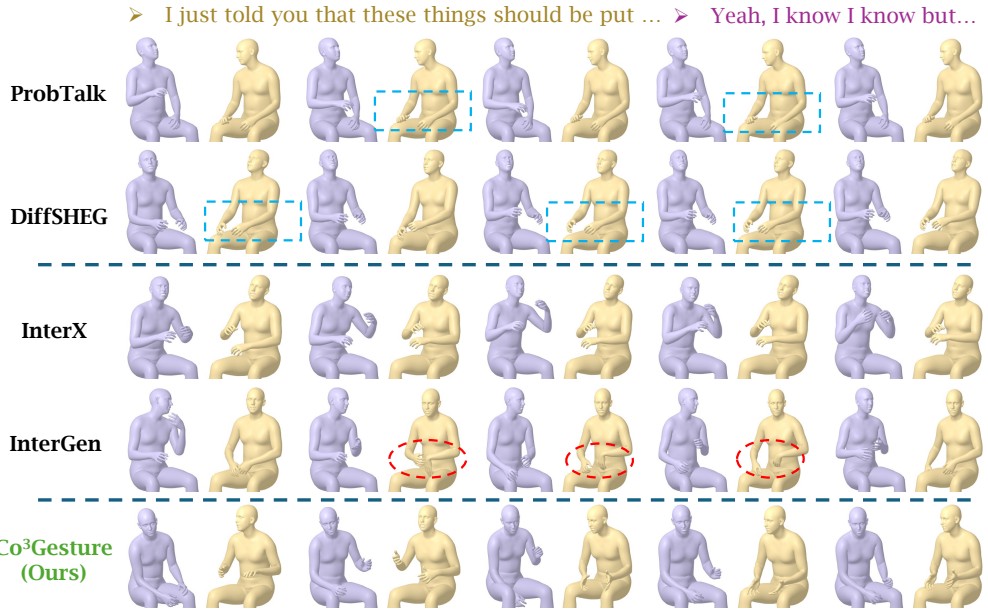

Figure 4: Visualization of our generated concurrent 3D co-speech gestures against various state-of-the-art methods. The samples are from our newly collected GES-Inter dataset.

are anonymously selected from various majors in school. For each method, we randomly select two generated videos in the user study. Hence, each participant needs to watch 16 videos for 6 seconds of each. The subjects are required to evaluate the generated results by all the counterparts in terms of naturalness, smoothness, and interaction coherency. The visualized videos are randomly selected and ensure that each method has at least two samples. The statistical results are shown in Figure 5 with the rating scale from 0-5 (the higher, the better).

Our framework demonstrates the best performance compared with all the competitors. To be specific, our method achieves noticeable advantages from the perspective of smoothness and interaction coherency. This indicates the effectiveness of our proposed bilateral denoising and temporal interaction module.

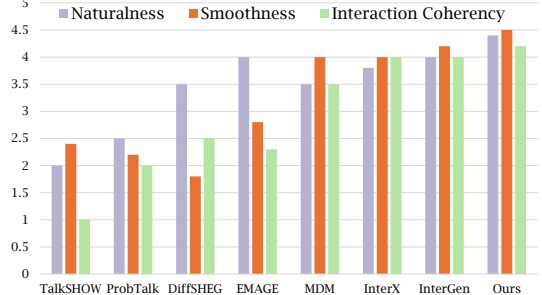

Figure 5: User study on gesture naturalness, motion smoothness, and interaction coherency.

## 5 CONCLUSION

In this paper, we introduce a new task of coherent concurrent co-speech gesture generation given conversational human speech. We first newly collected a large-scale dataset containing more than 7M concurrent co-speech gesture instances of two speakers, dubbed GES-Inter. This high-quality dataset supports our task while significantly facilitating the research on 3D human motion modeling. Moreover, we propose a novel framework named Co$^3$Gesture that includes a temporal-interaction module to ensure the generated gestures preserve interactive coherence. Extensive experiments conducted on our GES-Inter dataset show the superiority of our framework.

**Limitation.** Despite the huge effort we put into data preprocessing, the automatic pose extraction stream may influence our dataset with some bad instances. Meanwhile, our framework only generates the upper body movements without expressive facial components. In the future, we will incorporate our framework with tailor-designed facial expression modeling and investigate more stable data collection techniques to further improve the quality of our dataset. Besides, we will put more effort into designing specific interaction metrics for better concurrent gesture evaluation.

ACKNOWLEDGMENT

The research was supported by Early Career Scheme (ECS- HKUST22201322), Theme-based Research Scheme (T45- 205/21-N) from Hong Kong RGC, NSFC (No. 62206234), and Generative AI Research and Development Centre from InnoHK.

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

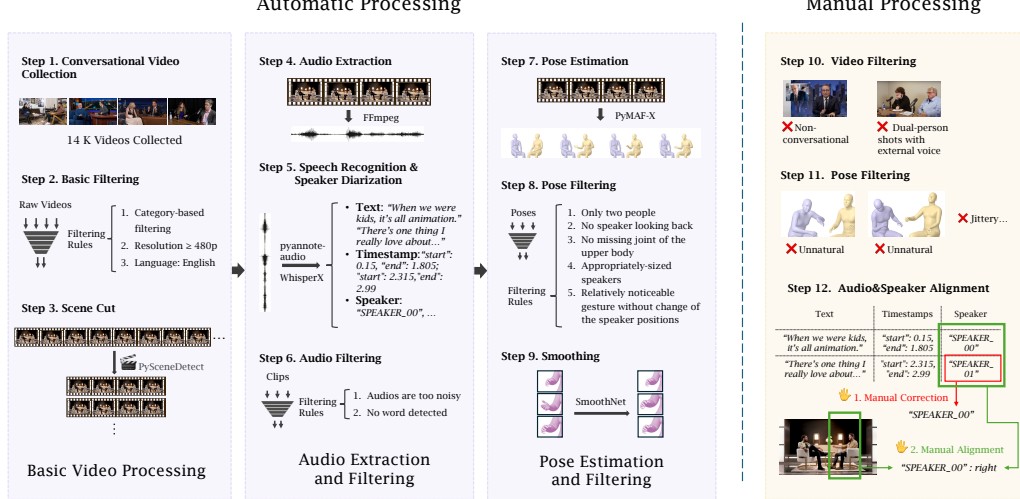

Figure 6: The overall workflow of our dataset construction. The videos are processed to obtain high-quality postures through advanced automatic technologies and professional expert proofreading.

## A APPENDIX

To showcase the superior quality of our GES-Inter dataset and the effectiveness of our proposed Co³Gesture , we provide additional details on data collection and further visualization results below.

### A.1 DATASET CONSTRUCTION

In this section, we give a detailed explanation of the data processing pipeline of our GES-Inter dataset. We summarize the acquisition, processing, and filtering of our GES-Inter dataset into two main procedures: automatic and manual processing steps, as illustrated in Figure 6.

#### A.1.1 AUTOMATIC PROCESSING STEPS

To build a high-quality 3D co-speech gesture dataset with concurrent and interactive body dynamics, we collect a considerable number of videos. They are then processed using automated methods to extract both audio and motion information.

**Basic Video Processing (Step 1, 2, 3):** First, with related searching keywords, we collect more than 14K conversational videos along with their metadata (*e.g.*, video length, frame resolution, audio sampling rate, *etc.* ). Those keywords include talk show, conversation, interview, *etc.* In this step, we acquire raw video data totaling up to 1,095 hours. However, many of these videos do not meet our requirements regarding category, quality, language, and other factors. Therefore, we filter them in step 2 to make sure: i) the speakers in the videos are real people rather than cartoon characters; ii) the videos meet an acceptable quality standard, featuring a resolution of at least 480p for clear visuals; and iii) only English conversations are included. In these preprocessing phases, due to the large amount of raw video collected and labor consumption, we conduct initial filtering using automatic techniques without manually checking each video. Specifically, we leverage YOLOv8 for human detection, discarding clips that do not show realistic people (eg, cartoon characters). Meta information provided by downloaded videos directly filters the English conversational corpus. Following the initial filtering, we proceed to step 3, where we use PySceneDetect to cut the videos into short clips.

**Audio Extraction and Filtering (Step 4, 5, 6):** Audios and poses are two necessary attributes for our GES-Inter dataset. In Step 4, we extract audio from the video clips using FFmpeg. In step 5, we initially employed the pyannote-audio technique for speaker diarization, configuring it for two speakers to accommodate two-person dialogues. The pyannote-audio tool assigns each speech segment to the appropriate speaker. Next, we utilize WhisperX Bain et al. (2023) for speech-to-text transcription. After transcription, we cluster the speakers based on the generated timestamps to better

organize the dialogue. With the extracted audio and speaker diarization, we filter out clips in Step 6 that either have relatively low audio quality or no word detected. This filtering process improves the efficiency of the subsequent pose estimation.

**Pose Estimation and Filtering (Step 7, 8, 9):** As an acknowledged 3D human representation standard, SMPL-X Pavlakos et al. (2019) is used to represent whole-body poses in various related tasks Jiang et al. (2023); Zhang et al. (2022); Liu et al. (2024b;a); Chen et al. (2024); Liang et al. (2024b). Accordingly, we employ the cutting-edge pose estimator PyMAF-X Zhang et al. (2023a) to extract high-quality 3D postures including body poses, subtle fingers, shapes, and expressions of the speakers. We then apply five criteria to filter the clips based on the pose annotation: *containing only two people, no speaker looking back, no missing joint of the upper body, appropriately-sized speakers, and relatively noticeable gesture without change of the speaker positions.* However, upon examining the visualized motions, we still observe that some temporal jittering within the movements is inevitable. To this end, we exploit SmoothNet Zeng et al. (2022) for temporal smoothing and jittery motion refinement in Step 9. The jittery effects are mostly caused by the blurring of speakers moving quickly in consecutive video frames. Due to the strict keyword selection in raw video crawling, our dataset rarely contains two speakers standing or walking around. If there are several clips including the aforementioned postures, we will filter them out to ensure our dataset maintains unified posture representation.

In particular, our manual review indicates that SmoothNet effectively generates cleaner and more reliable motion sequences while maintaining a diverse range of postures. However, due to the frequent extreme variations in camera angles, speaker poses, and lighting in talk show videos, some inaccuracies in pose estimations from PyMAF-X are unavoidable. Thus, inspired by Pavlakos et al. (2019), we translate the joint of arm postures as Euler angles with $x$, $y$, and $z$ order. Then, if the wrist poses exceed 150 degrees on any axis, or if the pose changes by more than 25 degrees between adjacent frames (at 15 fps), we discard these abnormal postures over a span of 90 frames.

### A.1.2 MANUAL PROCESSING STEPS

Building on the initial postures and audio obtained automatically, we introduce manual processing in this section to further refine the annotation.

**Basic Video Filtering (Step 10):** We observe that several undesired clips have passed through the initial filtering, including non-conversational scenarios and dual-person shots with external voices. To ensure that all videos meet our standards, we recruit two groups of inspectors to meticulously review and eliminate any that do not comply with the specified criteria. The results of each group are sampled and inspected to guarantee authority.

**Pose Filtering (Step 11):** We conduct a manual review of the processed clips at a consistent ratio of 5:1, selecting one clip from each group of five while adhering to the order of scenecut. This approach is valid, as there may exist overlap among adjacent clips. All clips are organized into five groups, and each group is assigned to an inspector for thorough evaluation. The inspectors assess the visualizations using SMPL-X parameters to determine whether the motion appears smooth, jittery, or unnatural. If any sequences are identified as jittery or unnatural, we discard the entire group of five clips from which the sample was taken.

Additionally, we eliminate instances where the speakers experience significant occlusion of their bodies during the interaction. This meticulous evaluation process greatly enhances the quality of our GES-Inter dataset.

**Audio&Speaker Alignment (Step 12):** We obtained speech separation results with an accuracy of 95%. Here we defined correct instances as those audio clips with accurate speech segmentation, correct text recognition, and accurate alignment. During our audio preprocessing, the audio is initially segmented by pyannote-audio technique to achieve 92% accuracy. Then, the accuracy of text recognized by WhisperX is 96%.

Once we obtain the separated audios, to ensure the identity consistency between the separated audio and body dynamics, we conduct audio-speaker alignment in this step. To be specific, professional human inspectors are recruited to manually execute this operation. Inspectors first check every video clip with its diarization to ensure the sentence-level speaker identities are correct and consistent within the clip. Then, inspectors assign the specific spatial position, *i.e.*, left or right, to the speaker identities

Table 6: Statistical results in User Study. $\pm$ denotes standard deviation.

| Comparison Methods | Naturalness | Smoothness | Interaction Coherency |
|---|---|---|---|
| TalkSHOW | $2^{\pm 0.1}$ | $2.4^{\pm 0.6}$ | $1^{\pm 0.1}$ |
| ProbTalk | $2.5^{\pm 0.3}$ | $2.2^{\pm 0.3}$ | $2^{\pm 0.2}$ |
| DiffSHEG | $3.5^{\pm 0.5}$ | $1.8^{\pm 0.3}$ | $2.5^{\pm 0.6}$ |
| EMAGE | $4^{\pm 0.4}$ | $2.8^{\pm 0.4}$ | $2.3^{\pm 0.5}$ |
| MDM | $3.5^{\pm 0.6}$ | $4^{\pm 0.3}$ | $3.5^{\pm 0.1}$ |
| InterX | $3.8^{\pm 0.4}$ | $4^{\pm 0.5}$ | $4^{\pm 0.3}$ |
| InterGen | $4^{\pm 0.5}$ | $4.2^{\pm 0.2}$ | $4^{\pm 0.2}$ |
| Ours | $4.4^{\pm 0.2}$ | $4.5^{\pm 0.1}$ | $4.2^{\pm 0.1}$ |

in the diarization. Using the alignment and the revised diarization with timestamps, inspectors separate each extracted audio into two distinct files and name them according to the corresponding speakers. To ensure the high fidelity of the alignment, the initially aligned audio-speaker pairs are double-checked by another group of inspectors. Meanwhile, the human inspectors would further check the rationality of segmentation and text recognition results from the perspective of human perception. In this step, we set two groups of inspectors for cross-validation to ensure the final alignment rate is 98%. In this manner, our GES-Inter dataset contains high-quality human postures with corresponding separated authority human speeches and multi-modality annotations. We provide examples of audio separation for better demonstration (refer to our webpage: *https://mattie-e.github.io/Co3/*).

## A.2 MORE DETAILS ABOUT EXPERIMENTAL SETTING

Due to the complex and variable positions of the two speakers of in-the-wild videos, we set the relative positions of the two speakers to fixed values. In the experiments, we only model the upper body dynamics of the two speakers. In particular, the joint order follows the convention of SMPL-X. During experiments, we follow the convention of (Liu et al., 2022a;b; 2024a) to resample FPS as 15. In our dataset, we retain all the metadata (*e.g.*, video frames, poses, facial expressions) within the original FPS (*i.e.*, 30) of talk show videos. We will release our full version data and pre-processing code, thereby researchers can obtain various FPS data according to their tasks.

## A.3 MORE DETAILS ABOUT USER STUDY

In the user study, all participating students are asked to evaluate each video without any indication of which model generated it. For fair comparison in user study, the demo videos are randomly selected. We count the motion fractions length of two speakers upon all the 16 demo videos. We adopt elbow joints as indicators to determine whether the motion occurs. Empirically, when the pose changes by more than 5 degrees between adjacent frames, we nominate the speakers who are moving now. Therefore, among 16 demo videos with 6 seconds, the average motion fraction lengths of the two speakers are 4.3 and 3.1 seconds.

A higher score reflects better quality, with 5 signifying that the video fully meets the audience's expectations, while 0 indicates that the video is completely unacceptable. To ensure fairness, each video is presented on a PowerPoint slide with a neutral background. Before participants see the generated results, we show several pseudo-annotated demos in our dataset as reference. All participants are required to watch the video at least once before they rate it. We invite participants in batches at different time periods within a week. Once all students have submitted their ratings anonymously, we collect them to calculate an average score. After completing the statistics, we randomly selected 60% of them to rate again two weeks later, and the results show that there is no obvious deviation.

We report the detailed mean and standard deviation for each method as shown in Table 6. Our method even achieves a 10% ((4.4-4.0)/4=10%) large marginal improvement over suboptimal InterGen in Naturalness. Meanwhile, our method displays a much lower standard deviation than InterX and InterGen. This indicates the much more stable performance of our method against competitors.

Table 7: Significance Analysis of User Study

| Comparison Methods | Naturalness | Smoothness | Interaction Coherency |
|---|---|---|---|
| TalkSHOW | 5.345 | 3.567 | 4.123 |
| ProbTalk | 3.789 | 5.001 | 3.456 |
| DiffSHEG | 4.789 | 2.654 | 5.299 |
| EMAGE | 3.214 | 5.120 | 4.789 |
| MDM | 3.789 | 4.567 | 2.987 |
| InterX | 3.001 | 3.456 | 2.148 |
| InterGen | 2.654 | 3.299 | 2.234 |

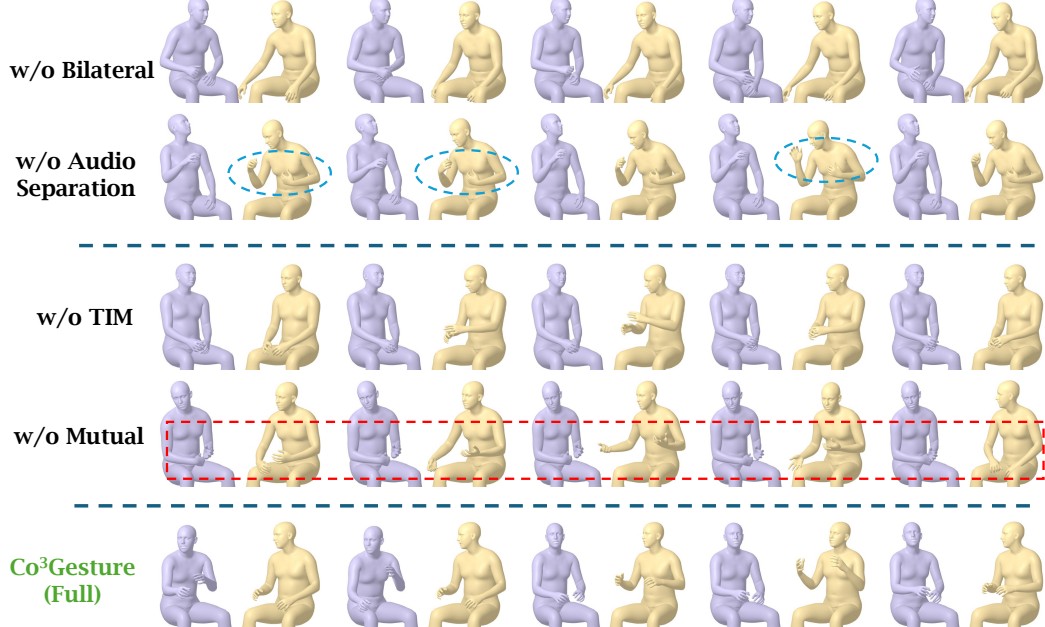

Figure 7: Visualization of our generated concurrent 3D co-speech gestures in the ablation study. Best view on screen.

To further verify the effectiveness of our user study, we conduct a significant analysis of the user study using t-test, focusing on three key aspects: Naturalness, Smoothness, and Interaction Coherency. The results verify our method surpasses all the counterparts with significant improvements, including sub-optimal InterGen. In particular, for all the comparisons between our model and the other models, we formulate our null hypotheses (H0) as "our model does not outperform another method". In contrast, the alternative hypothesis (H1) posits that "our model significantly outperforms another method," with a significance level ($\alpha$) set as 0.05. Here, we perform a series of t-tests to compare the rating scores of our model against each of the other competitors individually and calculate all the t-statistics shown in Table 7. Since we recruit 15 volunteers, our degree of freedom(df) for every analysis is 14. Then, we look up the t-table with two tails and find out all the p-values are less than 0.05 ($\alpha$). Therefore, we reject the null hypotheses, indicating that our model significantly outperforms all the other methods in every aspect.

## A.4 ADDITIONAL VISUALIZATION RESULTS

Here, we provide more visualization results of the ablation study in our experiments. As shown in Figure 7, the full version of our framework demonstrates the vivid and coherent interaction of body dynamics against other versions. We also display more visualized demo videos on our anonymous website: *https://mattie-e.github.io/Co3/*.

