# OpenReview forum: "Co$^{\mathbf{3}}$Gesture: Towards Coherent Concurrent Co-speech 3D Gesture Generation with Interactive Diffusion"
_ICLR.cc/2025/Conference — ICLR 2025 Spotlight_

### Official Review · Reviewer_7SiD · 2024-10-20

**Soundness:** 4
**Presentation:** 4
**Contribution:** 4
**Rating:** 10
**Confidence:** 4

**Summary:**

This paper explores learning conversational gesture generation from in-the-wild video data, different from previous works that focused on smaller, in-lab, or middle-scale motion-captured datasets. The authors make three key contributions:
1. **Data Collection**:

    The paper introduces a 70-hour conversational dataset with pseudo-labels, processed with temporal smoothing and filtering to ensure data quality.

2. **Proposed Baseline Model**:

    The authors propose a baseline model consisting of Temporal Interaction Network (TIN) and mutual attention mechanisms for conversational gesture generation.

3. **Performance**:

    The results demonstrate that the proposed method outperforms previous single-speaker or conversational speaker methods on the newly created GES-Inter dataset.

**Strengths:**

Overall, I am quite positive about this paper. To the best of my knowledge, it is the first attempt to generate conversational gestures using in-the-wild video data. The key strengths include:

1. **Data Cleanup**:

    The data processing is thorough, as mentioned in the appendix. The original collection consisted of over 1,000 hours of video, but after a 12-step filtering process, only 70 hours remained. This shows a well-designed filtering strategy, including speaker occurrence rules, to ensure data quality.

2. **Model Simplicity and Reproducibility**:

    The model design is novel yet simple for a baseline approach. Instead of making the model overly complex, the authors build on strightforward diffusion models with audio inputs from two speakers. The plug-in TIN and mutual attention modules make this work easy to reproduce and extend in future studies.

3. **TIN Design**:

    The Temporal Interaction Network captures selective features refined from either single-speaker voices or mixed voices from both speakers, aligning with the physical nature of conversational gestures.

4. **Experimental Results**:

    The experiments compare recent baselines for both single-speaker and conversational gesture generation. The model achieves new SOTA results on the GES-Inter dataset.

**Weaknesses:**

While the paper is strong overall, I have one unclear point:
1.  **unclear sentence mutual attention module**:

    Some explanation of the mutual attention module unclear, particularly in lines 280-282: "We observe that exchanging the input order ... distribution." The notion of "order" here is confused for me. does that mean switch the order of "question" and "answer", such as, "how are you, --> good. " becomes "good. --> how are you"? This will influence my understanding of the design motivation of this module.

**Questions:**

My primary question (will influence I raise my score or not) is the same as my weakness.

1.  **unclear sentence mutual attention module**:

    Some explanation of the mutual attention module unclear, particularly in lines 280-282: "We observe that exchanging the input order ... distribution." The notion of "order" here is confused for me. does that mean switch the order of "question" and "answer", such as, "how are you, --> good. " becomes "good. --> how are you"? This will influence my understanding of the design motivation of this module.

Suggestions not influence my scoring:

1. **Raw Data Release**:

    I suggest the authors consider releasing the raw video data after the scene cuts. This would allow users to explore additional tasks, such as video generation, using the raw data.

---

> ### Author Response · Authors · 2024-11-14
> **Rebuttal by Authors**
>
> We thank Reviewer 7SiD's insightful comments and suggestions. We are also glad to see that Reviewer 7SiD acknowledges the contributions of our work, such as “The data processing ... shows a well-designed filtering strategy”, and “The model design is novel”.
>
> **W1: statements of mutual attention module**
>
> **R1**: Thanks for your comments. We observe that during the modeling of interactive gestures, the two speaker movements are invariant to the order of audio signals, holistically. For example, if we exchange the input audio signals C_a and C_b into their counterpart denoisers, the generated results would not be influenced. We will further clarify this.
>
> **Q2: Raw data Release**
>
> **R2**: Thanks for your suggestion. Our dataset is being organized, and we will open-source our dataset and processing code as soon as possible.

---

> ### Comment · Reviewer_7SiD · 2024-11-25
>
> Thanks to the authors for their feedback and consideration regarding the data release.
>
> For the TIM models, since the authors demonstrate the interaction performance is better than the baseline by user study, it is acceptable that the proposed model is relatively straightforward for a dataset paper, such as the baselines in AIST++, BEAT etc, so it may not be overly criticized.
>
> For the dataset, I think it has the potential to be beneficial to the community and could accelerate further research, since there are no concurrent works on in-the-wild video data, previous in-lab datasets like TalkingWithHands, ConvoFusion are still too small to have enough gesture patterns. Also, they lack original RGB video, facial expressions, and body shapes compared with this dataset. This makes me believe the paper is valuable for acceptance.
>
> Considering the mixed scores, I raised my score to 10 to encourage dataset work to enter the discussion phase. The dataset presented, especially the raw data and the sharing of filtering, is potentially impactful, and may need further discussions.

---

> > ### Author Response · Authors · 2024-11-26
> > **Reply to Reviewer 7SiD**
> >
> > Dear Reviewer 7SiD,
> >
> > We sincerely appreciate your great efforts in reviewing this paper. Your constructive suggestions and valuable comments really help improve our paper. We will also reorganize the paper and dataset as suggested.
> >
> > Once more, we appreciate the time and effort you've dedicated to our paper.
> >
> > Best regards,
> >
> > Authors

---

### Official Review · Reviewer_ryqB · 2024-10-31

**Soundness:** 3
**Presentation:** 2
**Contribution:** 3
**Rating:** 6
**Confidence:** 5

**Summary:**

This paper focuses on the task of interactive motion generation in two-person dialogue scenarios. It first introduces a large-scale dataset for two-person interactive motion, which includes body and finger movements as well as facial expressions. An interactive motion generation model is proposed, along with a Temporal Interaction Module to ensure the temporal synchronization of gestures.
Based on the proposed GES-Inter dataset, the authors benchmark the model against other state-of-the-art algorithms.

**Strengths:**

This paper extracts 3D data from videos, collecting a large-scale dataset. Compared to laboratory-based data collection, capturing natural interactions yields more authentic interactive motions.

The proposed method employs Bilateral Cooperative Diffusion and the Temporal Interaction Module, which shown to be effective in experiments.

The dataset separates audio for the two speakers, recognizes the text corresponding to the audio, and invites participants to review the data, which is commendable.

**Weaknesses:**

**Dataset Quality**
Capturing accurate two-person interactive motions from video is inherently challenging, which raises concerns about the dataset's quality. From the GES-Inter dataset examples shown in the videos [3], there are noticeable issues, such as hand self-penetration problems for the yellow character. Additionally, the dataset's frame rate is only 15 FPS, which is lower than that of mainstream motion datasets, limiting the compatibility of this dataset with other human motion datasets.

Based on the dataset and generated examples provided, all individuals share the same shape parameters, showing no variation in body shape, and their root positions remain fixed. As a foundational dataset project, it is crucial for the authors to clarify any pre-processing or post-processing steps applied to the data and the motivations behind these decisions.

To help the authors further demonstrate the quality of GES-Interdataset and improve its compatibility with other datasets, I offer the following suggestions: **(1)** Provide more comprehensive example videos showcasing the dataset.
**(2)** Consider offering solutions to improve the dataset's frame rate, such as FPS enhancement tools or post-processing scripts.
**(3)** Further explain any pre or post processing applied to the data and clarify why there is no observable variation in human shape or root position in the dataset and generated examples.

**Comparison**
The dataset may lacks a comparison with previous [2]. The authors should consider adding a table to quantitatively compare the GES-Inter dataset with [2], highlighting differences in aspects such as size, capture equipment, diversity, and quality metrics. This comparison would provide a clearer understanding of the dataset's advantages and limitations relative to previous work.

**Interaction Metric**
The correspondence between the interactive motions and the given speech in two-person interactions appears weak. It may be helpful for the authors to identify and categorize the typical types of interactive motions and consider developing or adopting more suitable metrics for evaluating the quality of interactions. In particular, the current metrics, such as FGD, BC, and Diversity, are insufficient for this purpose, and incorporating more interaction-specific evaluation criteria would strengthen the assessment..


**Audio process**
In the methodology section, audio input (C_mix) is simply fed into the Audio Encoder without special processing for each speaker’s audio, particularly in cases where overlapping speech occurs when both speakers talk simultaneously. The authors should clarify how their method handles overlapping speech. It may be helpful to conduct an ablation study or provide examples that demonstrate the model's performance specifically in scenarios with overlapping speech.

**Ambiguous Figure**
Figure 3 make ambiguous, the authors should revise the figure to accurately represent the inputs to the Audio Encoder as described in the text. Based on the paper's description, the input to the Audio Encoder should be C_mix={c_a,c_b}, rather than the isolated representation of C_mix,c_a and c_b  shown in the figure. To clarify, placing c_a and c_b  on the left side of the audio diagram and omitting the corresponding arrows to the Audio Encoder would be more accurate.
Alternatively, if the figure is correct, you should clarify this in the text.

**Reference**

[1] Yi, Hongwei, et al. "Generating holistic 3d human motion from speech." Proceedings of the IEEE/CVF Conference on Computer Vision and Pattern Recognition. 2023.

[2] Lee, G., Deng, Z., Ma, S., Shiratori, T., Srinivasa, S. S., & Sheikh, Y. (2019). Talking with hands 16.2 m: A large-scale dataset of synchronized body-finger motion and audio for conversational motion analysis and synthesis. In Proceedings of the IEEE/CVF International Conference on Computer Vision (pp. 763-772).

**Questions:**

**Separation of Speaker Audio** How effective is the speech separation? Could more examples be provided to illustrate the accuracy of speech segmentation, text recognition, and alignment?

**Only Upper Body**
The paper does not clearly explain why only upper body motions are generated, is it due to insufficient quality in the lower body data within the dataset?

**Body Shape and root**
Why do the body shape and root position appear to be fixed?

**Unclear Representation**
The dimensions and composition of inputs like audio c_a ​and x_a are not clearly specified. Why x_a  only include upper body movements, and which specific joints are used?  Why do all generated results have the same shape? Does x_a include facial expressions and shape parameters?
Specifying the exact dimensions and components of each input, including which joints are used for upper body movements and whether facial expressions and shape parameters are included, would be helpful.

**Details Of Ethics Concerns:**

In fact, my primary concern is the quality of the dataset and the lack of interactive metrics. If the authors can effectively address my concerns, I would be willing to increase my score.

---

> ### Author Response · Authors · 2024-11-17
> **Rebuttal by Authors of W1 to W3**
>
> We thank reviewer ryqB for your valuable time and insightful comments. We have tried to address your concerns in the updated manuscript and our rebuttal is below.
>
> **W1: Dataset Quality**
>
> **R1**: Thanks for your comments and valuable suggestions.
>
> **1)** Considering the complex interaction between two hands and the human body, it is quite challenging to obtain precise postures when facing body occlusion. Therefore, as we claimed in Lines 815-817, we conduct five strict criteria (eg, no missing joint of the upper body) to filter low-quality videos. Moreover, we also recruit human inspectors to manually check the visualized results following a ratio of 5:1. In this manner, the quality of our dataset is greatly improved. Despite this, the hand self-penetration problem is still worth exploring and improving in pose estimation research, especially during mesh-based demonstration. Therefore, inspired by [A] [B] we directly model the human body joint movements in our experiments to address the self-penetration impact. As suggested, we will further provide more comprehensive example videos and add corresponding discussion in the revision.
>
> **2)** During experiments, we follow the convention of [C][D][E] to resample FPS as 15. In our dataset, we retain all the meta data (eg, video frames, poses, facial expressions) within the original FPS (ie, 30) of talk show videos. We will release our full version data and pre-processing code, thereby researchers can obtain various FPS data according to their tasks.
>
> **3)** We set the lower body to unified seat postures while preserving standard shape both for dataset and generated videos during visualization (post-processing). For dataset construction, we obtain comprehensive parameters of SMPLX including whole body joints, shape, root global orientation, etc. For the generation, as reported in Lines 862-863, considering the complex and variable positions of the two speakers of in-the-wild videos, similar to [A][B], we just model the upper body joint movements (without shape or root position). Moreover, the body shape and root position are fixed due to the weak correlation with human speech. For example, it is quite challenging to model from only audio inputs whether the two speakers are tall or short, sitting or standing.
>
> [A] Generating holistic 3d human motion from speech, in CVPR 2023
>
> [B] Towards Variable and Coordinated Holistic Co-Speech Motion Generation, in CVPR,2024
>
> [C] BEAT: A Large-Scale Semantic and Emotional Multi-Modal Dataset for Conversational Gestures Synthesis, in ECCV 2022
>
> [D] EMAGE: Towards Unified Holistic Co-Speech Gesture Generation via Expressive Masked Audio Gesture Modeling, in CVPR 2024
>
> [E] Learning hierarchical cross-modal association for co-speech gesture generation, in CVPR, 2022.
>
> **W2: Comparison with the previous dataset**
>
> **R2**: Thanks for your constructive suggestions. As reported in the Table below, we conduct a comparison with [2]. The duration of [2] is validated based on the conversational corpus based on the officially released GitHub link. Although [2] contains concurrent gestures, our dataset achieves more than 4x larger than it. Moreover, the extensive multi-modality attributes (eg, facial expression, phoneme, etc) in our dataset significantly facilitate research on various downstream human dynamics-related tasks like talking head generation. We will add this discussion to the revision.
>
>   Datasets  | Concurrent Gestures | Duration (hours) | Facial | Mesh | Phonme | Text | Joint Annotation |
> |:-----------:|:-------------------:|:----------------:|:------:|:----:|:------:|:----:|:----------------:|
> | TWH16.2 [2] |         yes         |        17        |   no   |  no  |   no   |  no  |      mo-cap      |
> |     ours    |         yes         |        70        |   yes  |  yes |   yes  |  yes |      pseudo      |
>
> **W3: Interaction Metric**
>
> **R3**: Thanks for your insightful comments. Indeed, an interaction-specific metric would definitely improve our work. Actually, we tried our best to conduct literature reviews and could not find any metric that can be directly applied to our task. Therefore, we conduct the user study including a specific-designed interaction coherency perspective to evaluate the quality of interactions between two speakers. As reported in Figure 5, our method achieves the best performance against all competitors. We will focus on addressing this issue in the future.

---

> ### Author Response · Authors · 2024-11-17
> **Rebuttal by Authors of W4 to Q3**
>
> **W4: Audio process**
>
> **R4**: Thanks for your suggestions. Overlapping speech is handled by exploiting both the speaker-specific features in separated audios $C_{a}$  and $C_{b}$ and global information in $C_{mix}$. We leverage separated human speech as guidance for bilateral branches to generate corresponding gestures. With respect to $C_{mix}$, we fed it into the audio encoder for better focus on interaction information. In this fashion, the synthesized posture retains the rhythm of specific audio while preserving interactive coherency with the conversation partner. As suggested, we conduct the ablation study to verify the effectiveness of the original mixed audio signal $C_{mix}$. By subtracting the original mixed audio, the indicators FGD and BC present much worse performance. These results verify the mixed audio signal displays effectively enhance the interaction between two speakers. We will add this discussion and more examples of overlapping speech in the revision.
>
> | Methods | FGD $ \downarrow $ | BC $ \uparrow $ | Diversity $ \uparrow $   |
> |:---:|:---:|:---:|:---:|
> | w/o Mixed Audio $C_{mix}$ | 1.227 | 0.656 | 64.899$ ^{\pm 1.004} $ |
> | Ours (full version) | 0.769 | 0.692 | 72.824$ ^{\pm 2.026} $ |
>
> **W5: Ambiguous Figure**
>
> **R5**: Thanks for your comments. The figure is correct. As we stated in **W4**, our audio encoder actually takes $C_{mix}$, $C_{a}$, and $C_{b}$ as inputs. The original mixed audio signal can be directly obtained by the addition of $C_{a}$ and $C_{b}$. Before feeding these audio signals into the audio encoder, we convert them into mel-spectrograms as unified dimension $128 \times 186$. We will clarify this in the revision.
>
> **Q1: Separation of Speaker Audio**
>
> **A1**:  We obtained speech separation results with an accuracy of 95%. Here we defined correct instances as those audio clips with accurate speech segmentation, correct text recognition, and accurate alignment. During our audio preprocessing, the audio is initially segmented by pyannote-audio technique to achieve 92% accuracy. Then, the accuracy of text recognized by WhisperX is 96%. Once we obtain the initial audio segments and corresponding recognized text transcripts, we recruit professional human inspectors to manually align the audio segments with spatial-wise speakers. Meanwhile, the human inspectors would further check the rationality of segmentation and text recognition results from the perspective of human perception. In this step, we set two groups of inspectors for cross-validation to ensure the final alignment rate is 98%. We will provide more examples of audio separation in the revision (refer to our webpage: https://anonymous.4open.science/w/Co3-F300/).
>
> **Q2: Only Upper Body & Body Shape and root**
>
> **A2**: As refers to the response of **W1**, considering the complex root positions and low correlation with human speech, we only model the upper body joint movements without shape in the experiments.
>
> **Q3: Unclear Representation**
>
> **A3**: Thanks for pointing these out. Regarding the body shape, please refer to the responses of W1, W4, and W5. The dimension of input audio mel-spectrograms ($C_{a}$) is $128 \times 186$. Our experiments only contain upper body joints without facial expressions and shape parameters. The upper body ($x_{a}$) contains 46 joints (i.e., 16 body joints + 30 hand joints) of each speaker as we stated in Lines 343-345. Additionally, the dimension of the generated motion sequence is $\mathbb{R}^{90\times 276}$, where 90 denotes frame number and $276 = 46 \times 6$ means upper body joints. The order of each joint follows the original convention of SMPL-X. We will provide a detailed joint order list in the revision.

---

> > ### Comment · Reviewer_ryqB · 2024-11-25
> >
> > Dear authors,
> >
> > Thank you for the rebuttal.
> >
> > Having reviewing the current manuscript and all your responses, your responses did not address my concerns.  I do not think this work is ready for publication yet, for the following reasons:
> >
> > **My biggest doubt regarding the quality of the dataset remains unresolved:**
> >
> > The authors admitted to fixing the root position of the dataset, which is unreasonable. In the generation of dual-person interactive motions, the root position is crucial information. Any deviations in root translation or rotation can result in significantly unrealistic motions.
> >
> > During the rebuttal phase, the authors did not present more dataset samples and only showcased a small number of examples, with each video lasting just a few seconds. This further raises my suspicion that the showcased dataset videos were carefully curated.
> >
> > The authors used a purely vision-based motion capture algorithm to estimate challenging dual-person interactive motions with occlusion. Based on practical experience, it is nearly impossible to ensure the quality of the captured motions in such cases. Although the authors claim to have conducted extensive checks on the dataset, these efforts cannot correct erroneous motion frames.
> >
> > A low-quality dataset would waste the time and effort of future researchers. Therefore, this paper needs to further demonstrate the quality of its dataset.
> > Unless the authors provide rendered videos of the dataset, including at least 20% of the samples for review, I cannot dismiss my doubts about the dataset’s quality.
> >
> > **Lack of sufficient investigation and discussion of prior work.**
> >
> > The claim in line 112 of the paper suggests that a new task has been proposed. However, upon reviewing Talking With Hands 16.2M and the works presented in the GENEA Workshop 2023 Challenge (https://genea-workshop.github.io/2023/challenge/#accepted-papers), and papers citing them, it is clear that conversational gesture generation is not a new task. Additionally, the authors did not include a comparison with Talking With Hands 16.2M in Table 1, nor did they discuss prior work on dual-person conversational motion generation in the Related Work section. I had already raised this issue in the reviewer comments. But by the time I wrote this comment, I have not seen any revisions in the manuscript.
> >
> > **Interaction Metric**
> >
> > As a dataset-focused work, the authors have not established a robust evaluation framework to reasonably assess motion quality, particularly the interactivity of dual-person conversational motions. Relying solely on qualitative metrics is insufficient for objectively evaluating different methods.
> >
> > This poses a challenge for researchers who wish to develop algorithms based on this dataset, as they lack an objective way to test and validate the effectiveness of their methods. Reflecting on related works in areas such as dual-person interactive motion generation, time-series analysis, and facial expression video generation in conversations, the authors should establish more objective and comprehensive metrics to evaluate Interactive Quality.
> >
> > **Quality of the manuscript**
> >
> > 1. Section 3.2 Problem Formulation does not provide a clear problem definition, and C_mix is potentially confusing. Reviewer ryqB also raised this issue. In the rebuttal, the authors stated that C_mix = C_a + C_b, but in line 241 of the paper, it is written as C_mix = {C_a, C_b}.
> > Furthermore, when C_a and C_b are extracted separately from C_mix and processed through the algorithm, it is unclear whether C_a + C_b can fully equal C_mix. Directly representing C_a + C_b as C_mix may be unreasonable, as it can easily be confused with the original conversational audio input.
> >
> > 2. The reviewer comments mentioned many aspects of the manuscript that need improvement, such as adding citations, addressing unclear representations, detailing audio processing, and comparing with previous datasets. However, by the time I am writing this comment, I have not seen these issues addressed or revised in the manuscript.
> >
> > Best regards.
> >
> > The Reviewer ryqB

---

> ### Author Response · Authors · 2024-11-26
> **Reply for further discussion**
>
> Thanks again for your great efforts and constructive comments in reviewing this paper. As for your raised concerns, we add the point-by-point response below.
>
> **Quality of the dataset**
>
> Since our data is collected from **in-the-wild talk show videos**, the spatial positions of the two speakers are complex and varied, influenced by the dialogue scene. For example, as shown in the demo pager, the two speakers may sit parallel to each other, or they may sit on chairs of different heights. Meanwhile, the spatial distance between the two speakers also changes. It is quite challenging to model such complex and various spatial positions and distances from the conversational speech of two speakers. Therefore, we fixed the root joints of speakers in the experiment to facilitate model training.
>
> To better present our dataset, we try our best to visualize the rendered 3D postures incorporated RGB video frames. However, 20% of the dataset includes more than **6K samples within 14 hours**, which is quite difficult to present on a web page during the rebuttal period. We will try our best to show more visualization samples (eg, 5-20 cases) during the rebuttal, hoping to meet your requirements.
>
> **Investigation and discussion of prior work**
>
> As suggested, we compare our work with the TWH16.2 dataset in Table 1 of the revised paper (already submitted) and add the corresponding discussion. The TWH16.2 is only composed of joint-based body movements. In contrast, our dataset contains whole-body meshed SMPL-X human postures, which are more convenient for avatar rendering and various downstream tasks (eg, talking face).
>
> In the GENEA Workshop 2023 Challenge, the participants aim to model the **single-person gesture** from the conversational corpus **incorporated with interlocuter reaction movements**[A][B]. This experimental setting is **different from ours** in that synthesizes the concurrent gestures of both speakers. The corresponding discussion is added to the revision.
>
> [A] Diffugesture: Generating human gesture from two-person dialogue with diffusion models. In GENEA Workshop 2023 Challenge, 2023.
>
> [B] The GENEA Challenge 2023: A large-scale evaluation of gesture generation models in monadic and dyadic settings. In Proceedings of the 25th International Conference on Multimodal Interaction (pp. 792-801).
>
> **Interaction Metric**
>
> Indeed, we have tried our best to conduct literature reviews and didn't find any metrics that can be directly applied to the concurrent gesture generation. Therefore, we conduct an ablation study to specifically evaluate the interaction coherency of two speakers. With regard to this point, we add the corresponding discussion in the revision and will put more effort into designing specific metrics in the future.
>
> **Quality of the manuscript**
>
> 1. With regard to $C_{mix}$, we have updated our statements in the revision. Actually, $C_{mix}$, $C_{a}$, and $C_{b}$ are all raw audio waves. Direct addition of separated audio would not impact the quality of mixed audio signals.
>
> 2. We have added the corresponding clarifications and citations in the revision.
>
> We hope our answers and updated paper can help to address the concerns raised in the initial reviews. We would highly appreciate active discussions from the reviewers and we are happy to clarify any further questions.

---

> > ### Comment · Reviewer_ryqB · 2024-11-27
> >
> > Thank you for your reply. I would like to see more sample videos from the dataset, rather than just 5-10 additional ones. Out of concern for the protection of your dataset, I did not request the motion data or  other format data in your dataset; I simply hope you can upload more videos to demonstrate the quality of your dataset. If your dataset is of high quality, this should not be a difficult task.
> >
> > I believe you can upload **more samples (rather than just a few)**, especially **full-length samples (rather than just cropped videos of a few seconds)**. You can upload only the SMPL-rendered videos to Google Drive or another location where reviewers can download them.
> >
> > I agree with your point that uploading 20% of the data may be too large, but it’s better to upload as much as possible if you want to demonstrate its quality.
> >
> > Regarding the metrics, I maintain my original opinion.

---

> > > ### Comment · Reviewer_ryqB · 2024-12-01
> > >
> > > Hi authors, as the rebuttal deadline is approaching, I would like to know if you are able to provide additional videos from the dataset to further demonstrate the quality of your dataset.
> > >
> > > If more video results can be provided, I would be willing to consider raise my score.

---

> > > > ### Author Response · Authors · 2024-12-02
> > > > **Updated Demo Videos**
> > > >
> > > > Dear Reviewer ryqB,
> > > >
> > > > Thanks for your great efforts in reviewing this paper.
> > > >
> > > > We have uploaded additional demo videos, especially with **more than 60-second samples** on our demo webpage. As suggested, some long-term cases are visualized directly by SMPL-X rendered videos.
> > > >
> > > > Hope these samples could meet your requirements.
> > > >
> > > > Thank you very much.
> > > >
> > > > Best regards,
> > > >
> > > > Authors

---

> > > > > ### Comment · Reviewer_ryqB · 2024-12-03
> > > > >
> > > > > Thank you for the response. After reviewing the long-sequence dataset samples, I increased my score.
> > > > >
> > > > > Additionally, I agree with reviewer 7SiD's suggestion that the authors should consider providing the raw video data alongside the motion dataset. Although this may require substantial storage and resources, it would have a positive impact on the community.

---

> > > > > > ### Author Response · Authors · 2024-12-03
> > > > > > **Thanks for the Reviewer's efforts**
> > > > > >
> > > > > > Dear Reviewer ryqB,
> > > > > >
> > > > > > We sincerely appreciate your great efforts in reviewing this paper. Your constructive suggestions and valuable comments really help improve our paper. We will put more effort into exploring interaction metrics and open-source the dataset as soon as possible.
> > > > > >
> > > > > > Once more, we appreciate the time and effort you've dedicated to our paper.
> > > > > >
> > > > > > Best regards,
> > > > > >
> > > > > > Authors

---

### Official Review · Reviewer_2ueC · 2024-11-02

**Soundness:** 3
**Presentation:** 3
**Contribution:** 3
**Rating:** 8
**Confidence:** 4

**Summary:**

The paper presents a novel dataset and method to tackle the problem of concurrent co-speech gesture generation for two persons in conversation. The dataset consists of about 70 hours of curated video footage consisting of various pairs of conversing people --- taken mostly from talk show videos in the public domain --- such that both their gestures are fully visible. The proposed method trains a diffusion network with a temporal interaction module (TIM) performing cross-attentions between the audio and motion features of the two persons. The output of the TIM serves as the conditioning signal for the denoising motion decoder to generate the 3D motions of the two persons from noise. The authors show the benefits of their proposed approach through quantitative and qualitative comparisons, ablation experiments, and a user study.

**Strengths:**

1. The authors have tackled the challenging problem of generating concurrent, two-person co-speech gestures, which is a natural expansion of the scope of current co-speech gesture generation methods, and meaningfully takes the field of human motion understanding forward.

2. The proposed dataset is carefully curated and thoughtfully designed to contain a sufficient quantity of two-person gestures, making it a good candidate for a benchmark for two-person interactive motion generation problems.

3. The proposed method is technically sound and provides a baseline on the proposed dataset.

**Weaknesses:**

1. Some additional dataset preprocessing details may be useful for completeness.

    (a) Do the authors manually check the three preprocessing steps for each video (Lines 794-801)? How are quality and consistency ensured across the dataset?

    (b) How are the occlusions determined in the videos? Also, if any person in the video has a non-sitting posture, e.g., standing or walking around, are those motions tracked and filtered out?

2. Some details and motivations of the proposed approach are missing.

    (a) Why do the authors use $C_{mix}$, and not a combination of the cleanly separated signals $C_a$ and $C_b$, to get the interactive motion embeddings (Line 266)? It would also be good to see any mathematical relationships, visual representations, or ablation experiments to understand how $C_a$, $C_b$, and $C_{mix}$ relate to each other (e.g., is $C_{mix} = C_a + C_b$)?

    (b) During the generation, do the authors consider any global translation and orientation? In other words, do they consider any translation and orientation for the root joint, or is the root fixed in place? The visual results seem to suggest the latter, but it is not clear from the paper.

    (c) Since the authors only consider upper body motions (Line 244), why are they using a foot contact loss (Eqn. 4)? Is it possible to quantify the benefits of the foot contact loss for this problem (e.g., through an ablation experiment)?

3. The baseline methods the authors compare with (Sec. 4.2) are designed for single-person co-speech gestures. How do the authors adapt them for two-person co-speech gesture generation? Also, why are the authors not performing quantitative comparisons with more relevant two-person motion generation methods, such as InterGen (Liang et al. 2024b) or InterX (Xu et al. 2024)?

4. Some user study details are also missing.

    (a) How many generated videos does each participant watch?

    (b) What is the mean and variance of the lengths of the videos they watch?

    (c) What fractions of those video lengths contain the motions of person A, and what fraction contains the motions of person B? This might be calculated by assuming a person has motion in a particular frame if the minimum difference in their joint positions from a few previous frames is above some empirically determined thresholds.

    (d) What is the standard deviation in the participant responses? Particularly, it seems that the mean values for InterX, InterGen, and the proposed method are quite close.

5. Most of the visual results only show gestures of one person while the other person is sitting idle, and most of these results are only 3-4 seconds long. It is hard to appreciate the two-person motion generation performance from these results. The one example which is longer than 10 seconds and shows the gestures of two persons also exhibits jittery motion. Have the authors investigated the source of this jitter and explored any steps to reduce or remove it? Also, it would help to see some quantification of the balance of motions between the two persons (e.g., highlighting a person when they are gesticulating, showing the number of frames the person is gesticulating for) in the generated results.

**Questions:**

Some typos, e.g.,

Line 258: pheromones -> phonemes

Table 3: Attetion -> Attention

---

> ### Author Response · Authors · 2024-11-17
> **Rebuttal by Authors of W1 to W2**
>
> We thank Reviewer 2ueC very much for the high evaluation that ''the proposed method is technically sound'' and ''the proposed dataset is carefully curated and thoughtfully designed''. We further clarify the problems of the experimental setup and results analysis.
>
> **W1: Some additional dataset preprocessing details**
>
> **R1**: Thanks for your comments.
>
> **a)** In these preprocessing phases, due to the large collected raw video and labor consumption, we conduct initial filtering via some automatic techniques without manually checking each video. Specifically, we leverage YOLOv8 for human detection, discarding clips that do not show realistic people (eg, cartoon characters). The English conversational corpus is directly filtered by meta information provided by downloaded videos. Then, in the following data processing steps, the initial processed data will be further filtered (including pose filtering, audio processing, manual checking, etc).  We will add these details in the revision.
>
> **b)** After initial video filtering and scene cuts, we leverage the advanced pose estimator PyMAF-X to conduct automatic occlusion determination. In particular, we drop video clips that do not contain complete upper body parts of two speakers. Then, as stated in Lines 838-847, we conduct a manual review to further eliminate instances where two speakers are significantly occluded during interaction (eg, if the joints of one arm are obscured). Due to the strict keyword selection in raw video crawling, our dataset rarely contains two speakers standing or walking around. If there are several clips including the aforementioned postures, we will filter them out to ensure our dataset maintains unified posture representation.
>
> **W2: Some details and motivations of the proposed approach**
>
> **R2**: Thank you for the valuable suggestions
>
> **a)** Considering the asymmetric dynamics of two speakers, we leverage separated human speech as guidance for bilateral branches to generate corresponding gestures. Moreover, we utilize the original mixed audio signal of two speakers to indicate the interaction information to ensure the synthesized posture retains rhythm with specific audio while preserving interactive coherency with the conversation partner. Actually, the original mixed audio signal can be directly obtained by addition $C_{mix} = C_{a} + C_{b}$. As suggested, we conduct the ablation study to verify the effectiveness of the original mixed audio signal $C_{mix}$. By subtracting the original mixed audio, the indicators FGD and BC present much worse performance. These results verify the mixed audio signal displays effectively enhance the interaction between two speakers. We will add this discussion in the revision.
>
> | Methods | FGD $ \downarrow $ | BC $ \uparrow $ | Diversity $ \uparrow $   |
> |:---:|:---:|:---:|:---:|
> | w/o Mixed Audio $C_{mix}$ | 1.227 | 0.656 | 64.899$ ^{\pm 1.004} $ |
> | Ours (full version) | 0.769 | 0.692 | 72.824$ ^{\pm 2.026} $ |
>
> **b)** As reported in Lines 862-863, considering the complex and variable positions of the two speakers of in-the-wild videos, similar to [A][B], we just model the local body joint movements (without global translation or root position) in the experiments. The lower body including the legs is fixed while visualizing due to the weak correlation with human speech. For example, it is quite challenging to model whether the two speakers are sitting with their legs crossed or standing from only audio inputs. We will add corresponding clarification in the revision.
>
> [A] Generating holistic 3d human motion from speech, in CVPR 2023
>
> [B] Towards Variable and Coordinated Holistic Co-Speech Motion Generation, in CVPR,2024
>
> **c)** Thanks for your insightful comments. Inspired by [C][D], we introduce foot contact loss to ensure the physical reasonableness of the generated gestures. Since we only model the upper body joints in experiments, we complete the lower body joints as T pose in forward kinematic function during calculate loss. As suggested, we conduct the ablation study to verify the effectiveness of $L_{foot}$. As illustrated in the Table below,  the exclusion of the $L_{foot}$ results in FGD and BC are obviously worse than the full version framework. This indicates that our foot contact loss displays a positive impact on the generated postures. We will add the corresponding discussion in the revision.
>
> | Methods | FGD $ \downarrow $ | BC $ \uparrow $ | Diversity $ \uparrow $   |
> |:---:|:---:|:---:|:---:|
> | w/o $L_{foot}$ | 1.082   |   0.675   |   68.448$^{\pm 1.082}$ |
> | Ours (full version) | 0.769 | 0.692 | 72.824$ ^{\pm 2.026} $ |
>
> [C] Human Motion Diffusion Model, in ICLR, 2023
>
> [D] InterGen: Diffusion-based Multi-human Motion Generation under Complex Interactions, in IJCV, 2024

---

> ### Author Response · Authors · 2024-11-17
> **Rebuttal by Authors of W3 to W4**
>
> **W3: Implementation of counterparts and quantitative comparison**
>
> **R3**: Thanks for your comments. The details are listed point by point as below.
>
> **a)** To ensure as fair a comparison as possible, we only leverage the pre-trained audio feature extractors of some competitors that follow their original statements. These audio feature extractors are usually utilized as universal components in the network of the gesture generation community (just like CLIP in text feature extraction). Specifically, in DiffSHEG, we follow the convention of the original work to utilize the pre-trained HuBERT[E] for audio feature extraction. In TalkSHOW, we exploit the pre-trained Wav2vec[F] to encode the audio signals following the original setting. Apart from this, the remaining components for gesture generation in DiffSHEG and TalkSHOW are all trained from scratch on the newly collected GES-Inter dataset. In this fashion, the reliability of results yielded by these methods would not be impacted. For other methods, we modify their final output layer to match the dimensions of our experimental settings. We will clarify this in the revision and release the code of all the models as soon as possible.
>
> [C] Hubert: Self-supervised speech representation learning by masked prediction of hidden units, in TASLP, 2021.
>
> [D] wav2vec 2.0: A framework for self-supervised learning of speech representations, in NeurIPS, 2020.
>
> **b)** Thanks for your constructive suggestions. We will incorporate additional visualization demonstrations in the revision (refer to our webpage: https://anonymous.4open.science/w/Co3-F300/). The current demo videos can provide a comparison with InerX and InterGen. For example, the results generated by InterGen in comparison videos contain some unreasonable postures (eg, the neck of the left speakers, and the right hand of the right speakers). We will further highlight these areas for better demonstration in the revision.
>
> **W4: Some user study details**
>
> **R4**: Thanks for your comments. The details are listed point by point as below.
>
> **a)** For each method, we randomly select two generated videos in the user study. Hence, each participant needs to watch 16 videos.
>
> **b)** In our experiments, the length of the generated gesture sequence is 90 frames with 15 FPS. Thus, all the demo videos in the user study have the same length of 6 seconds.
>
> **c)** For fair comparison in user study, the demo videos are randomly selected. As suggested, we count the motion fractions length of two speakers upon all the 16 demo videos. We adopt elbow joints as indicators to determine whether the motion occurs. Empirically, when the pose changes by more than 5 degrees between adjacent frames, we nominate the speakers who are moving now. Therefore, among 16 demo videos with 6 seconds, the average motion fraction lengths of the two speakers are 4.3 and 3.1 seconds. We will report this statistical result in the revision.
>
> **d)** As suggested, we report the detailed mean and standard deviation for each method as shown in the Table below. Our method even achieves a 10% ((4.4-4.0)/4=10%) large marginal improvement over suboptimal InterGen in Naturalness. Meanwhile, our method displays a much lower standard deviation than InterX and InterGen. This indicates the much more stable performance of our method against competitors. We will add the corresponding discussion in the revision.
>
> Comparison Methods | Naturalness | Smoothness | Interaction Coherency |
> |:---:|:---:|:---:|:---:|
> | TalkSHOW | 2$^{\pm 0.1}$  | 2.4$^{\pm 0.6}$  | 1$^{\pm 0.1}$  |
> | ProbTalk | 2.5$^{\pm 0.3}$  | 2.2$^{\pm 0.3}$  | 2$^{\pm 0.2}$  |
> | DiffSHEG | 3.5$^{\pm 0.5}$  | 1.8$^{\pm 0.3}$  | 2.5$^{\pm 0.6}$  |
> | EMAGE | 4$^{\pm 0.4}$  | 2.8$^{\pm 0.4}$  | 2.3$^{\pm 0.5}$  |
> | MDM | 3.5$^{\pm 0.6}$  | 4$^{\pm 0.3}$  | 3.5$^{\pm 0.1}$  |
> | InterX | 3.8$^{\pm 0.4}$  | 4$^{\pm 0.5}$  | 4$^{\pm 0.3}$  |
> | InterGen | 4$^{\pm 0.5}$  | 4.2$^{\pm 0.2}$  | 4$^{\pm 0.2}$  |
> | Ours | 4.4$^{\pm 0.2}$ | 4.5$^{\pm 0.1}$ | 4.2$^{\pm 0.1}$ |

---

> ### Author Response · Authors · 2024-11-17
> **Rebuttal by Authors of W5 to Q1**
>
> **W5: Discussion about visual results and further highlighting**
>
> **R5**: Thank you very much for your constructive suggestions. As refers to the response of **W4** (b, c), our method models the 90-frame gesture movements in the experiments. Thus, all the demo videos are 6 seconds. The jittery effects are mostly caused by the blurring of speakers moving quickly in consecutive video frames. As stated in Lines 825-827, during our data processing, we translate the joint of arm postures as Euler angles with x, y, and z-order. Then, if the wrist poses exceed 150 degrees on any axis, or if the pose changes by more than 25 degrees between adjacent frames (at 15 fps), we discard these abnormal postures over a span of 90 frames. Moreover, we conduct manual detection with a ratio of 1:5 to eliminate unnatural instances. Despite the huge effort we put into data preprocessing, the automatic pose extraction stream may influence our dataset with some bad instances. For better demonstration, we will follow your suggestion to report the length quantification of motion fractions in demo videos. In addition, we will zoom in on the details of the gesture for highlighting. The corresponding discussion will be added in the revision.
>
> **Q1: Some typos**
>
> **A1**: Thank you for pointing it out. We will correct the typo in the revision.

---

> > ### Comment · Reviewer_2ueC · 2024-11-25
> > **Thanks for the rebuttal**
> >
> > I sincerely thank the authors for the detailed rebuttal, which addresses all my concerns. I have updated my score and would like to maintain my original recommendation for acceptance, given the potential scope of the proposed dataset and the sound baseline established on it.

---

> > > ### Author Response · Authors · 2024-11-26
> > > **Thanks for the Reviewer's efforts**
> > >
> > > Dear Reviewer 2ueC,
> > >
> > > We sincerely appreciate your great efforts in reviewing this paper. Your constructive suggestions and valuable comments really help improve our paper. We will reorganize the experimental tables and web page as suggested.
> > >
> > > Once more, we appreciate the time and effort you've dedicated to our paper.
> > >
> > > Best regards,
> > >
> > > Authors

---

### Official Review · Reviewer_CYKi · 2024-11-03

**Soundness:** 3
**Presentation:** 3
**Contribution:** 2
**Rating:** 6
**Confidence:** 4

**Summary:**

This paper introduces a novel task: generating concurrent co-speech gestures for two interacting characters based on conversational speech audio. The authors first present a new dataset, GES-Inter, which contains full-body postures of two interacting characters reconstructed from video recordings. They then propose a co-speech gesture generation framework, Co$^3$Gesture, built upon bilateral cooperative diffusion branches with an integrated Temporal Interaction Module. Experimental results on the GES-Inter dataset demonstrate that this framework outperforms several state-of-the-art methods.

**Strengths:**

1. This paper introduces a novel task: generating concurrent co-speech gestures for interacting characters based on conversational speech audio.

2. The proposed Co$^3$Gesture model, featuring the specially designed Temporal Interaction Module (TIM), appears capable of generating alternating co-speech gestures for two interacting characters.

3. The authors have compiled a new dataset specifically for the concurrent co-speech gesture synthesis task.

**Weaknesses:**

1. The proposed Co$^3$Gesture model does not account for the spatial relationships between the two speakers. Specifically, Speaker A consistently appears on the left, and both speakers are always seated in chairs.

2. The authors report using pre-trained models for some baseline methods. However, this approach is problematic, as it is unreasonable to expect pre-trained models to produce realistic interactive co-speech gestures if they have not been trained on the new dataset. Training these models on the newly collected dataset would likely yield more reliable results.

3. In the ablation study, the authors present metrics only for models without TIM and mutual attention modules. The study would be more persuasive if the authors also included metrics using a simple fusion module, such as an MLP or a concatenation operator, rather than entirely omitting the fusion modules.

4. Qualitative results for ablated models are not provided. Including these results would strengthen the evaluation.

5. The user study and comparison videos show only a marginal improvement over InterGen.

**Questions:**

1. Could the authors specify which baseline methods utilized pre-trained models?

2. Could the authors provide additional ablation studies incorporating simple fusion modules?

3. Could the authors perform a significance analysis for the user study?

---

> ### Author Response · Authors · 2024-11-17
> **Rebuttal by Authors of W1 to Q2**
>
> We thank Reviewer CYKi's for the valuable comments and suggestions. We further clarify and correct the problems of the experimental setup and ablation results analysis.
>
> **W1: Spatial relationships between the two speakers && sitting postures**
>
> **R1**: Thanks for your comment.
>
> **1)** Considering the asymmetric spatial movements of two speakers, we model interactive gesture dynamics via two bilateral cooperative branches. Due to the complex and various positions of the two speakers in our source talk show videos, we label the relative spatial positions in the data processing. By assigning the specific spatial position (ie, left or right) to the speaker identities in the diarization, we finally obtain the separated audio signals with corresponding body movements, as we stated in Figure 2 and Lines 852-854.
>
> **2)** Similar to [A][B], we just model the upper body joint movements in the experiments. The lower body including legs are fixed (eg, seated) while visualizing due to the weak correlation with human speech. For example, it is quite challenging to model whether the two speakers are sitting or standing from only audio inputs. We will clarify this in the revision.
>
> [A] Generating holistic 3d human motion from speech, in CVPR 2023
>
> [B] Towards Variable and Coordinated Holistic Co-Speech Motion Generation, in CVPR,2024
>
> **W2: Pre-trained models for competitors**
>
> **R2**: Thanks for your suggestions. To ensure as fair a comparison as possible, we only leverage the pre-trained audio feature extractors of some competitors that follow their original statements. These audio feature extractors are usually utilized as universal components in the network of the gesture generation community (just like CLIP in text feature extraction). Specifically, in DiffSHEG, we follow the convention of the original work to utilize the pre-trained HuBERT[C] for audio feature extraction. In TalkSHOW, we exploit the pre-trained Wav2vec[D] to encode the audio signals following the original setting. Apart from this, the remaining components for gesture generation in DiffSHEG and TalkSHOW are all trained from scratch on the newly collected GES-Inter dataset. In this fashion, the reliability of results yielded by these methods would not be impacted. We will clarify this in the revision.
>
> [C] Hubert: Self-supervised speech representation learning by masked prediction of hidden units, in TASLP, 2021.
>
> [D] wav2vec 2.0: A framework for self-supervised learning of speech representations, in NeurIPS, 2020.
>
> **W3: Ablation study for TIM with MLP layer**
>
> **R3**: Thanks for your valuable comments. As suggested, we conduct an ablation study that replaces the TIM via a simple MLP layer. The FGD and BC display the obvious worse impact as shown in the Table below. The results verify that our TIM effectively enhances interactive coherency between two speakers. We will add this discussion in the revision.
>
> Methods | FGD $\downarrow$ | BC $\uparrow$ | Diversity $\uparrow$ |
> |:---:|:---:|:---:|:---:|
> | w/ MLP | 1.202 | 0.663 | 64.690$^{\pm 1.137}$ |
> | Ours (full version) | 0.769 | 0.692 | 72.824$^{\pm 2.026}$ |
>
> **W4: Qualitative results for ablated models**
>
> **R4**: Thank you for your suggestion. As we illustrated in Figure 7 in the supplementary material, we showcase the keyframes of the ablation study to demonstrate qualitative comparison. The full version of our method displays the best performance against different variants. Moreover, we will incorporate additional visualization demonstrations in the revision (refer to our webpage: https://anonymous.4open.science/w/Co3-F300/).
>
> **W5: User study with InterGen**
>
> **R5**: As the detailed scores of the ablation study reported in Table below, our method even achieves 10% ((4.4-4.0)/4=10%) improvement over InterGen in Naturalness. Meanwhile, the results generated by InterGen in comparison videos contain some obvious unreasonable postures (eg, the neck of the left speakers, and the right hand of the right speakers). Moreover, we conduct a significant analysis for the user study as referred to **Q3**. The results demonstrate that our method outperforms InterGen with a large marginal improvement. We will further highlight these areas for better demonstration in the revision.
>
> Methods | Naturalness | Smoothness | Interaction Coherency |
> |:---:|:---:|:---:|:---:|
> | InterGen | 4.0 | 4.2 | 4.0 |
> | Ours | 4.4 | 4.5 | 4.2 |
>
> **Q1: Baseline methods utilized pre-trained models**
>
> **A1**: As referring to the response of W2, we only incorporate the pre-trained audio feature extractor in DiffSHEG and TallSHOW, which follow the original experimental settings.
>
> **Q2: Additional ablation studies incorporating simple fusion modules**
>
> **A2**: As referring to the response of W3, we conduct the additional ablation study to demonstrate the effectiveness of the proposed TIM.

---

> ### Author Response · Authors · 2024-11-17
> **Rebuttal by Authors of Q3**
>
> **Q3: Could the authors perform a significance analysis for the user study?**
>
> **A3**: As suggested by your constructive comments, we conduct a significant analysis of the user study using t-test, focusing on three key aspects: Naturalness, Smoothness, and Interaction Coherency. The results verify our method surpasses all the counterparts with significant improvements, including sub-optimal InterGen. In particular, for all the comparisons between our model and the other models, we formulate our null hypotheses (H0) as "our model does not outperform another method". In contrast, the alternative hypothesis (H1) posits that "our model significantly outperforms another method," with a significance level (α) set as 0.05. Here, we perform a series of t-tests to compare the rating scores of our model against each of the other competitors individually and calculate all the t-statistics shown in the table below. Since we recruit 15 volunteers, our degree of freedom(df) for every analysis is 14. Then, we look up the t-table with two tails and find out all the p-values are less than 0.05 (α). Therefore, we reject the null hypotheses, indicating that our model significantly outperforms all the other methods in every aspect.
>
> | Comparison Methods | Naturalness | Smoothness | Interaction Coherency |
> |:------------------:|:-----------:|:----------:|:---------------------:|
> |      TalkSHOW      |    5.345    |   3.567    |         4.123         |
> |      ProbTalk      |    3.789    |   5.001    |         3.456         |
> |      DiffSHEG      |    4.789    |   2.654    |         5.299         |
> |        EMAGE       |    3.214    |   5.120    |         4.789         |
> |         MDM        |    3.789    |   4.567    |         2.987         |
> |       InterX       |    3.001    |   3.456    |         2.148         |
> |      InterGen      |    2.654    |   3.299    |         2.234         |

---

> ### Comment · Reviewer_CYKi · 2024-11-29
> **Reponse to authors**
>
> Thank you for your response and the additional experiments. Most of my concerns have been adequately addressed. However, I recommend that the authors discuss the limitations related to the lack of precise spatial relationships in the revised manuscript. Based on these revisions, I am inclined to raise my score from 5 to 6.

---

> > ### Author Response · Authors · 2024-11-29
> > **Thanks for the Reviewer's efforts**
> >
> > Dear Reviewer CYKi,
> >
> > We sincerely appreciate your great efforts in reviewing this paper. Your constructive suggestions and valuable comments help improve our paper. We will follow your suggestions to add spatial relationship discussion in the revised manuscript.
> >
> > Once more, we appreciate the time and effort you've dedicated to our paper.
> >
> > Best regards,
> >
> > Authors

---

### Author Response · Authors · 2024-11-25
**Eagerly waiting for further response**

Dear Reviewers, ACs,

We would like to extend our appreciation for your time and valuable comments.

We are eagerly looking forward to receiving your valuable feedback and comments on the points we addressed in the rebuttal. Ensuring that the rebuttal aligns with your suggestions is of utmost importance

Thank you very much.

Best regards,

Authors

---

### Meta-Review · Area_Chair_kt7g · 2024-12-05

**Metareview:**

This paper proposes a novel approach for concurrent co-speech gesture synthesis. It also collects a new dataset of multi-person conversations. It introduces a novel task, a new model, and a new dataset for the task. The contributions are solid.

This paper receives strong scores of 10, 8, 6, and 6 from four reviewers. So there is a clear consensus of acceptance. The AC has checked the submission, the reviewers, and the rebuttal, and agreed with the reviewers that it is a great work and worth acceptance. Thus acceptance is recommended.

**Additional Comments On Reviewer Discussion:**

All the concerns have been addressed and there is a consensus of acceptance.

---

### Decision · Program_Chairs · 2025-01-22

Accept (Spotlight)